# Targeting TREM2 signaling shows limited impact on cerebrovascular calcification

Sucheta Sridhar[1,2] , Yingyue Zhou[3] , Adiljan Ibrahim[4], Sergio Bertazzo[5] , Tania Wyss[6] , Amanda Swain[3], Upasana Maheshwari[1] , Sheng-Fu Huang[1] , Marco Colonna[3] , Annika Keller[1,2]

**Brain calcification, the ectopic mineral deposits of calcium phosphate, is a frequent radiological finding and a diagnostic criterion for primary familial brain calcification. We previously showed that microglia curtail the growth of small vessel calcification via the triggering receptor expressed in myeloid 2 (TREM2) in the *Pdgfb^{ret/ret}* mouse model of primary familial brain calcification. Because boosting TREM2 function using activating antibodies has been shown to be beneficial in other disease conditions by aiding in microglial clearance of diverse pathologies, we investigated whether administration of a TREM2-activating antibody could mitigate vascular calcification in *Pdgfb^{ret/ret}* mice. Single-nucleus RNA-sequencing analysis showed that calcification-associated microglia share transcriptional similarities to disease-associated microglia and exhibited activated TREM2 and TGFβ signaling. Administration of a TREM2-activating antibody increased TREM2-dependent microglial deposition of cathepsin K, a collagen-degrading protease, onto calcifications. However, this did not ameliorate the calcification load or alter the mineral composition and the microglial phenotype around calcification. We therefore conclude that targeting microglia with TREM2 agonistic antibodies is insufficient to demineralize and clear vascular calcifications.**

## Introduction

Brain calcification, whether vascular or parenchymal, is a frequent finding on radiological scans in older adults and in various brain pathologies, including tumors, inflammation, Alzheimer's disease (AD), and microgliopathies (Maheshwari et al, 2022). These radiological features are also characteristic of certain pediatric conditions, and the pattern of calcification can aid in their diagnosis (Goncalves et al, 2020a; 2020b). Similarly, in the case of primary familial brain calcification (PFBC), the presence of bilateral calcifications in specific brain regions is a key diagnostic criterion (Balck

et al, 2021). Autosomal dominant PFBC is a genetic neurodegenerative disorder caused by mutations in four genes: *XPR1*, *SLC20A2*, *PDGFRB*, and *PDGFB* (Wang et al, 2012; Keller et al, 2013; Nicolas et al, 2013; Legati et al, 2015). Clinically affected patients present with either motor symptoms such as bradykinesia and ataxia, or non-motor symptoms such as mood imbalances, anxiety, and cognitive deficit (Balck et al, 2021). However, although the presence of bilateral basal ganglion calcification is a diagnostic criterion for PFBC, the role of calcifications in disease initiation and progression is still unknown. Although a few reports have investigated bisphosphonates as a potential treatment to halt calcification and improve symptoms in PFBC (Loeb, 1998; Loeb et al, 2006; Oliveira & Oliveira, 2016), the therapeutic efficacy of halting calcification in this disease remains unclear.

Vascular calcification in PFBC patients and animal models (e.g., *Pdgfb^{ret/ret}*, *Xpr1^{+/−}*, *Slc20a2^{−/−}*) is accompanied by conspicuous microglial reactivity (Miklossy et al, 2005; Keller et al, 2013; Zarb et al, 2019; Nahar et al, 2020; Maheshwari et al, 2023). Although this observation could simply reflect a reactive response to the pathological change, our studies using the *Pdgfb^{ret/ret}* mouse model of PFBC revealed that reducing microglial numbers using an inhibitor (PLX5622) against colony-stimulating factor 1 receptor in *Pdgfb^{ret/ret}* mice significantly increased the calcification load (Zarb et al, 2021). This evidence suggests that microglia are not merely reactive, but instead play a crucial role in limiting the growth and progression of vascular calcifications in PFBC. In fact, microgliopathies with cell-autonomous microglial dysfunction lead to diseases that are accompanied by brain calcification (Bianchin & Snow, 2022), highlighting the role of proper microglial function in preventing ectopic calcification. In addition, mice without microglia display aging pathologies including demyelination (McNamara et al, 2023) and, most notably, vascular and parenchymal calcification in the thalamus (Chadarevian et al, 2024; Munro et al, 2024). Microglia have emerged as disease modulators in late-onset Alzheimer's disease, with key risk allelic variants being exclusively or highly expressed in microglia (Villegas-Llerena et al, 2016). Interestingly, a mouse model

---

[1]Department of Neurosurgery, Clinical Neuroscience Center, University Hospital Zurich, University of Zurich, Zurich, Switzerland    [2]Neuroscience Centre Zurich, University of Zurich and ETH Zurich, Zurich, Switzerland    [3]Department of Pathology and Immunology, Washington University School of Medicine, St Louis, MO, USA    [4]Alector, South San Francisco, CA, USA    [5]Department of Medical Physics and Biomedical Engineering, University College London, London, UK    [6]TDS-facility, AGORA Cancer Research Center, Swiss Institute of Bioinformatics, Lausanne, Switzerland

Correspondence: annika.keller@usz.ch

of AD lacking microglia developed parenchymal calcifications (Kiani Shabestari et al, 2022). Thus, these findings collectively suggest a link between microglial function and brain calcification in various disease contexts and preclinical animal models. These data also indicate that microglia are active participants in regulating brain calcification, potentially opening new avenues for therapeutic interventions in reducing brain calcification.

The molecular mechanisms by which microglia orchestrate the removal or control of ectopic calcification remain elusive. However, our previous studies suggest a critical role of the TREM2 in halting vascular calcifications. Deleting *Trem2* in the *Pdgfb^{ret/ret}* PFBC model significantly worsened calcification (Zarb et al, 2021). Notably, patients with loss-of-function mutations in *TREM2* also present with brain calcification (Paloneva et al, 2002; Bianchin et al, 2004). This indicates that TREM2 signaling plays a key role in preventing and potentially reversing brain calcification. TREM2 is known to regulate key microglial functions such as phagocytosis, inflammation, and migration (Jay et al, 2017). Because of the multifaceted role of TREM2 in microglial physiology, enhancing TREM2 function is emerging as a promising therapeutic approach for late-onset Alzheimer's disease and other neurodegenerative diseases: potentially by simultaneously improving phagocytic capacity and metabolism of microglia (Schlepckow et al, 2023).

In this study, we characterized the transcriptomic profile and activated signaling pathways of calcification-associated microglia (CAM) in a mouse model of PFBC (*Pdgfb^{ret/ret}*). We showed that CAM share similarities in their transcriptional profile to disease-associated microglia (DAM), and exhibit activated TREM2 and TGFβ signaling. We further investigated the possibility of halting vascular calcification growth in *Pdgfb^{ret/ret}* mice by boosting TREM2 activity to enhance microglial function by administering a bivalent anti-TREM2–activating antibody (AL002a). Our findings demonstrate that sustained administration of an anti-TREM2–activating antibody increased microglial deposition of cathepsin K on calcifications, aligning with our previous observation of its TREM2 dependence. However, we did not observe a reduction in the overall vascular calcification load. Therefore, although TREM2 activity in microglia appears to be necessary for controlling vascular calcification in *Pdgfb^{ret/ret}* mice, stimulating TREM2 alone as a therapeutic approach is insufficient to reduce it. This suggests that the processes by which microglia control tissue calcification are complex and likely involve interplay with other signaling pathways.

# Results

### Transcriptional profile of CAM

Our previous analysis of CAM reported their protein and cell surface profiles based on a limited number of selected markers (Zarb et al, 2021). To obtain an unbiased profile of their phenotype, we performed single-nucleus (sn) RNA sequencing (RNA-seq) of *Pdgfb^{ret/ret}* mouse brains to characterize the transcriptome of CAM. Nuclei were isolated from two different regions of control and mutant (*Pdgfb^{ret/ret}*) mice: the cerebral cortex, where vessels are not calcified and CAM are absent, and the deep brain, where blood

vessels are calcified and CAM are present (Fig S1A). A total of 119,935 nuclei from all deep brain samples clustered into 10 groups (Fig S1B). The clusters were identified based on the expression of brain cell–type markers (Zeisel et al, 2018; Zhou et al, 2020) as follows: neurons (clusters N1 to N5, expressing *Rbfox3*), oligodendrocytes (OC, expressing *Mbp*, *Mog*, *Plp1*), astrocytes (AC, expressing *Slc1a2*, *Aqp4*, *Aldh1l1*), and oligodendrocyte precursor cells (OPC, expressing *Pdgfra*, *Cspg4*, *Olig1*). The vascular cluster (VC) contained nuclei from mural cells (expressing *Vtn*, *Pdgfrb*) and endothelial cells (expressing *Cldn5*) (Vanlandewijck et al, 2018; Zhou et al, 2020). Microglia and perivascular macrophage nuclei were in the MG cluster and expressed markers *P2ry12*, *Csf1r*, and *Mrc1* (Fig S1C). Microglia from the MG cluster were analyzed further after removing contaminant nuclei that expressed non-microglial markers (i.e., *Mrc1*, *Mbp*, *Aqp4*, *Rbfox3*). A subsequent unsupervised clustering of 1,349 control (*Pdgfb^{ret/wt}*) and 1728 mutant (*Pdgfb^{ret/ret}*) microglial nuclei identified five separate clusters (Fig 1A). Two microglial clusters were determined as homeostatic (hMG1 and hMG2) based on the expression of known homeostatic microglial genes such as *P2ry12*, *Fcrls*, *Cx3cr1*, and *Siglech* (Keren-Shaul et al, 2017) (Fig 1B). The three remaining clusters were assigned as IFNR—interferon response microglia; PrMG—proliferating microglia; and CAM (Fig 1A). The IFNR cluster expressed interferon response genes such as *Nlrc5*, *Stat1*, and *Herc6*. The PrMG cluster up-regulated *Pola1* and *Cenpp*, indicating that the cluster contained microglia in various stages of the cell cycle. The CAM cluster was defined by the expression of genes such as *Apoe*, *Lpl*, *Axl*, *Spp1*, *Cd74*, *Csf1* and was mainly composed of nuclei from *Pdgfb^{ret/ret}* mice (Fig 1A and B). Analysis of pooled cortical and deep brain microglia showed that the CAM signature was only detected in deep brain microglia (Fig S1D and E). 96% of deep brain microglia from control mice clustered into hMG1 and hMG2 groups, in contrast to 89% of microglial nuclei from *Pdgfb^{ret/ret}* mice (Fig 1C). CAM were found to express *Spp1* transcripts (Figs 1B and D and S1F); however, expression at the protein level was not detected (Fig S1F).

Differential gene expression analysis of the CAM cluster as compared to the two homeostatic microglial clusters showed that CAM down-regulated homeostatic markers such as *P2ry12*, *Selplg*, *Siglech*, and *Tmem119* and up-regulated several genes including *Lpl*, *Flt1*, *Cd74*, *Axl*, *Itgax*, *Csf1*, *Spp1*, and *Apoe* (Fig 1D). To explore the functional roles of the up-regulated genes, a gene ontology analysis was performed. Up-regulated genes were involved in immune response, cell migration, and regulation of cell motility (Fig S1G). CAM also up-regulated phagocytic receptor *Axl* (Lemke, 2013) and vascular endothelial growth factor receptor *Flt1* (Fig S1G), which has been shown to play a role in microglial chemotactic response (Ryu et al, 2009). The up-regulation of pathways regulating cell motility and cytoskeletal changes (e.g., *Myo1e*, *Myo5a*, *Fmn1*) with up-regulation of phagocytosis receptors (e.g., *Axl*) indicated that the CAM are likely phagocytic (Figs 1D and S1G).

### CAM share transcriptomic signature with DAM

Microglia from diverse diseased brains share a core transcriptional signature indicative of a primed and responsive state, adopted in response to various insults (Krasemann et al, 2017). Although our

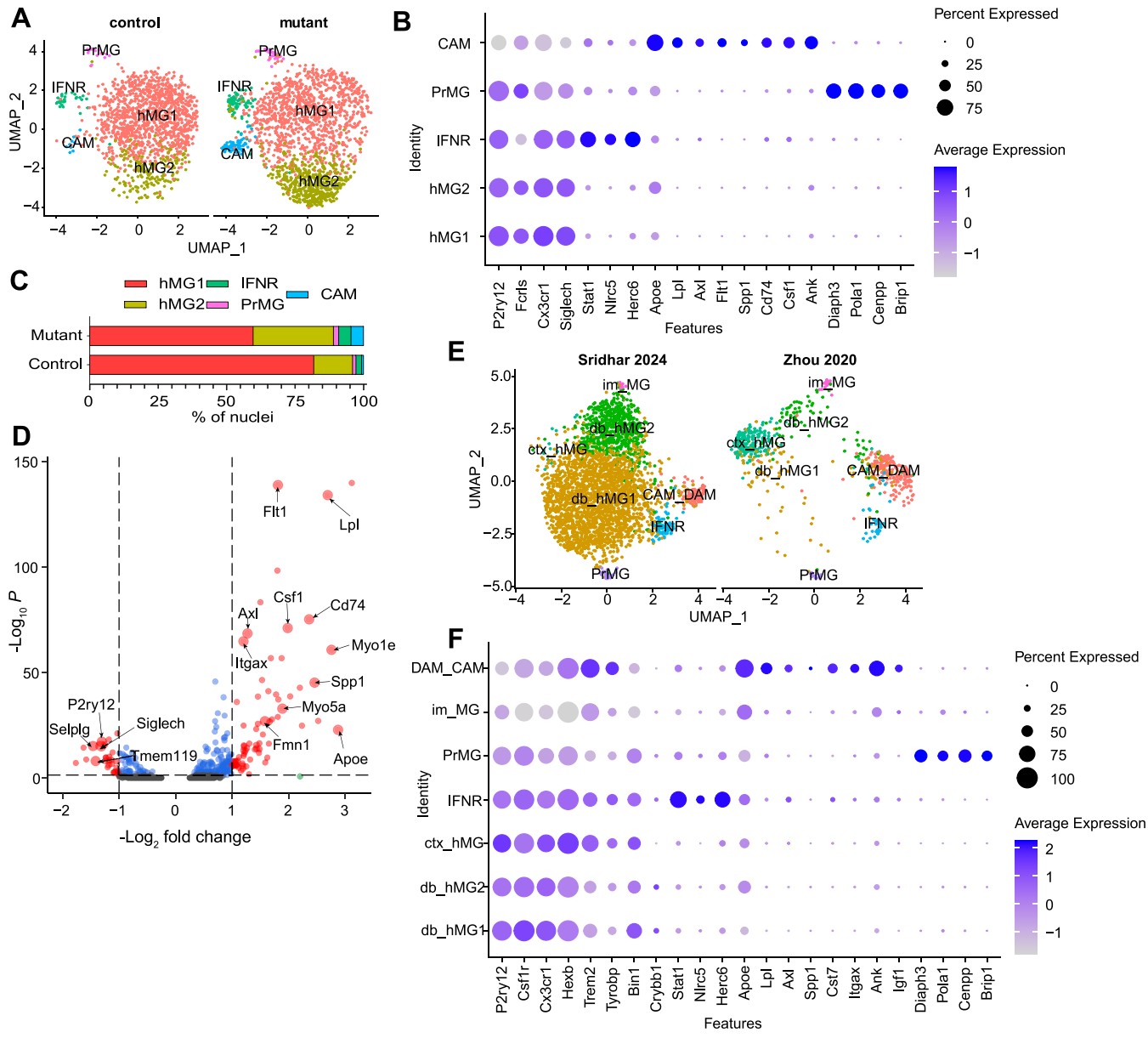

**Figure 1. Molecular signature of calcification-associated microglia.**
**(A)** Uniform Manifold Approximation and Projection of microglial nuclei from 5-mo-old mice (control [$Pdgfb^{ret/wt}$, n = 5]—1,349 nuclei; and mutant [$Pdgfb^{ret/ret}$, n = 6]—1,728 nuclei) from the deep brain region grouped into five clusters at a resolution parameter of 0.4. The individual clusters—hMG1, hMG2, IFNR, CAM, and PrMG—are separated by genotype. Age of mice: 5 mo. **(B)** Dot plots display the expression levels of cluster-identifying genes (features) by each cluster. The size of the dot represents the fraction of nuclei that express the gene, and the color reflects the scaled average level of expression. **(C)** Stacked bar plot depicting the percentage of nuclei in each microglial cluster and separated by genotype. **(D)** Volcano plot showing differentially expressed genes in CAM as compared to hMG1 and hMG2. Genes with log$_2$ fold change threshold > 1.0 and adjusted $P$ < 0.01 (Bonferroni correction) are labeled in red. **(E)** Uniform Manifold Approximation and Projection of deep brain microglial nuclei from this study integrated with the cortical microglial nuclei from 7-mo-old 5X-FAD and control mouse dataset (Zhou et al, 2020). Microglia grouped into six clusters—homeostatic MG from deep brain: db_hMG1 and db_hMG2; homeostatic MG from cortex: ctx_hMG, IFNR, PrMG, CAM_DAM, and im_MG. **(F)** Dot plots display the expression levels of cluster-identifying genes (features) by each cluster in the integrated analysis. The size of the dot represents the fraction of nuclei that express the gene, and the color reflects the scaled average level of expression. hMG, homeostatic microglia; IFNR, interferon response microglia; CAM, calcification-associated microglia; PrMG, proliferating microglia; im_MG, immature microglia.
Source data are available for this figure.

prior analysis using a limited set of markers suggested CAM resemble DAM of AD mouse models (Zarb et al, 2021), we aimed to determine whether this resemblance extends beyond those markers or indicates a broader overlap in their transcriptional

signatures. To address this, we integrated deep brain microglial nuclei from our dataset and cortical microglial nuclei from the Zhou et al snRNA-seq dataset (Zhou et al, 2020) and compared the transcriptomes of CAM and DAM. The homeostatic microglial

clusters separated between the two datasets (cortex versus deep brain) (Fig 1E), which is in line with previous observations that microglia show spatial heterogeneity (Tan et al, 2020). Homeostatic microglia (ctx_hMG) from the Zhou dataset expressed higher levels of *Hexb*, *Trem2*, and *Tyrobp* (Fig 1F), characteristic of cortical microglia (Tan et al, 2020). The homeostatic microglia from our dataset were enriched in clusters deep brain (db)_hMG1 and db_hMG2, whereas the homeostatic microglia from the Zhou dataset were enriched in cluster cortex (ctx)_hMG (Fig 1F). IFNR and PrMG clusters were identified in both datasets (Fig 1E and F). We also identified a small population of immature microglia (im_MG) (Fig S1H) (Hammond et al, 2019). The CAM clustered together with DAM (CAM_DAM cluster) (Fig 1E). This cluster up-regulated both DAM and CAM marker genes (Fig 1F) *Apoe*, *Lpl*, *Axl*, *Spp1*, *Cst7*, *Itgax*, *Ank*, and *Igf1* (Keren-Shaul et al, 2017) and down-regulated homeostatic genes *P2ry12*, *Csf1r*, and *Cx3cr1*, showing that the two states share a broad transcriptional signature (Fig 1E and F). Thus, CAM closely resemble DAM in their core transcriptional signature.

### TREM2 and TGFβ signaling is activated in CAM

Our previous work demonstrated TREM2 haploinsufficiency and knockout exacerbated calcification pathology in *Pdgfb*$^{ret/ret}$ mice (Zarb et al, 2021), indicating that the TREM2 signaling pathway was crucial for microglial regulation of calcification. However, unlike for DAM where the loss of TREM2 leads to loss of the DAM signature (Keren-Shaul et al, 2017; Zhou et al, 2020), CAM retained their phenotype in *Pdgfb*$^{ret/ret}$ mice (Zarb et al, 2021). Therefore, to clarify TREM2 signaling activity in CAM and homeostatic microglia, we used phosphorylated DAP12 (pDAP12) as a proxy for active TREM2 signaling. We quantified immunofluorescence of pDAP12 signal in IBA1-positive microglia in the calcified area and IBA1 and CLEC7A (CAM marker) double-positive CAM surrounding calcifications (Fig 2A). CAM showed significantly higher pDAP12 intensity compared with other microglia (MG) in the same region in *Pdgfb*$^{ret/ret}$ mice (Fig 2B), suggesting that TREM2 signaling is up-regulated in CAM.

In response to triggers such as Aβ plaques and dying neurons, it was shown that microglia acquire the more responsive DAM state in a TREM2-dependent manner by suppressing homeostatic signaling pathways such as TGFβ signaling via SMAD3 (Krasemann et al, 2017). Because the snRNA-seq data analysis of CAM and DAM showed a broad transcriptional similarity (Fig 1E and F) and CAM have activated TREM2 signaling (Fig 2A and B), we investigated whether CAM also show down-regulated TGFβ signaling in *Pdgfb*$^{ret/ret}$ mice. To assess TGFβ signaling, we measured the intensity of phosphorylated nuclear SMAD3 (pSMAD3), a downstream effector that, when phosphorylated, translocates to the nucleus and activates gene expression (Massague & Wotton, 2000). We quantified immunofluorescence of nuclear pSMAD3 signal in IBA1-positive microglia in the calcified area and IBA1- and CLEC7A-positive CAM surrounding calcifications (Fig 2C). The pSMAD3 signal was increased in the nuclei of CAM as compared to microglia from the same area (Fig 2D). Thus, although CAM share a transcriptomic signature with DAM, they differ in the signaling pathways activated in a responsive state.

### Administration of AL002a increases cathepsin K in vascular calcifications but does not reduce the overall calcification load

Because we observed an up-regulation of TREM2 signaling in CAM and a worsened vascular calcification pathology in *Trem2*-deficient *Pdgfb*$^{ret/ret}$ mice, we explored whether the antibody-mediated activation of TREM2 might be beneficial in reducing the vascular calcification burden. To address this, we used a murine anti-TREM2 antibody (AL002a) with an intact Fc region that binds the stalk region of the extracellular domain of TREM2 and activates the TREM2 signaling pathway (Price et al, 2020). An immunoassay with soluble TREM2 to capture the anti-TREM2 antibody was used for antibody quantification, and thus, only AL002a and not isotype antibody was detected (Fig 3A). The level of AL002a in the brains of control mice was similar to the amount reported before (Cignarella et al, 2020); however, a 100-fold higher level of AL002a antibody was found in the brains of *Pdgfb*$^{ret/ret}$ mice compared with control mice (Fig 3A). The increased blood–brain barrier permeability to IgG in *Pdgfb*$^{ret/ret}$ mice (Armulik et al, 2010) likely explains the higher level of AL002a. To assess the brain distribution of AL002a or isotype control, we administered fluorescently labeled AL002a or isotype control to *Pdgfb*$^{ret/ret}$ mice. Both AL002a and isotype control were distributed throughout the *Pdgfb*$^{ret/ret}$ brains, but cortical regions displayed stronger fluorescence intensity (Fig 3B). This is in line with the increased blood–brain barrier permeability of cortical regions in *Pdgfb*$^{ret/ret}$ mice as compared to deeper brain regions (Vanlandewijck et al, 2015; Keller et al, 2018). Both AL002a-AF647 and isotype-AF647 colocalized with IBA1-positive microglia. Colocalization of isotype-AF647 with microglia is likely indicative of interactions between the isotypes with Fc receptors expressed in microglia. However, only AL002a-AF647 and not isotype-AF647 coincident microglia presented with the strong pDAP12 signal (Fig 3C). Thus, AL002a reaches the brain and engages with microglia in *Pdgfb*$^{ret/ret}$ mice.

We next administered either AL002a or isotype control weekly for 8 wk, starting at 2 mo of age when vascular calcifications can already be observed in *Pdgfb*$^{ret/ret}$ mice (Keller et al, 2013). Vascular calcifications in PFBC patients and mouse models increase in number, size, and anatomical distribution over time (Keller et al, 2013; Zarb et al, 2019; Nahar et al, 2020; Balck et al, 2021). We terminated the treatment at 4 mo of age, a stage where we had previously mapped the regional pattern and calcification load in *Pdgfb*$^{ret/ret}$ mice (Zarb et al, 2019) (Fig S2A). Our previous data showed that CAM express and deposit cathepsin K in vascular calcifications in a TREM2-dependent manner (Zarb et al, 2021). We therefore asked whether activating TREM2 with AL002a would enhance cathepsin K deposition in vascular calcifications of *Pdgfb*$^{ret/ret}$ mice. Using anti-osteopontin staining, a bone matrix protein deposited in ectopic calcifications (Zarb et al, 2019), we visualized calcifications in these mice and quantified the cathepsin K signal within calcifications (Fig 3D). Compared with isotype control–treated mice, AL002a treatment led to an overall increase in cathepsin K deposition within the calcifications. Notably, isotype-treated mice exhibited greater inter-individual variability in their cathepsin K signal (Fig 3E). In bone, cathepsin K is expressed and secreted by osteoclasts and degrades the mineralization-permissive extracellular matrix (Bromme et al, 1996;

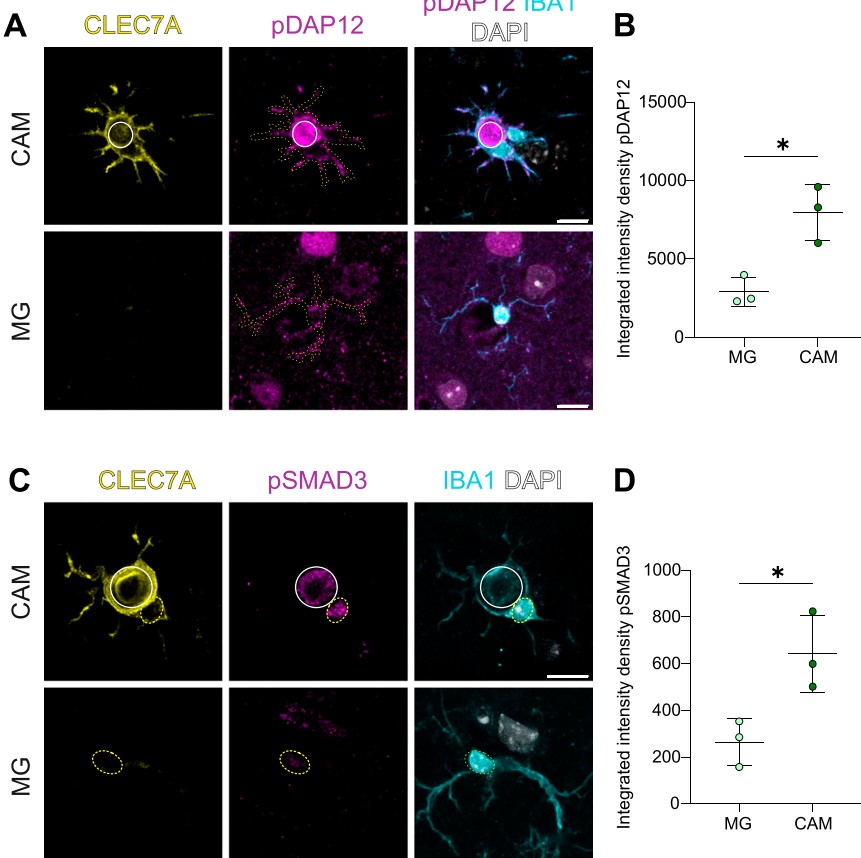

**Figure 2. TREM2 and TGFβ signaling is up-regulated in CAM in vivo.**
**(A)** Calcification-associated microglia (CAM), identified by CLEC7A (yellow) and IBA1 (cyan) around calcification (white circle), up-regulate pDAP12 (magenta), within microglial cell body depicted with yellow dots as compared to other microglia (MG) in the same region (lower panel). **(B)** Quantification of pDAP12 intensity between CAM and MG in the same region in 4-mo-old *Pdgfb*^*ret/ret*^ mice (n = 3 mice, unpaired two-tailed *t* test, *P* = 0.0127). **(C)** CAM, identified by CLEC7A (yellow) and IBA1 (cyan) around calcification (white dotted circle), have increased pSMAD3 signal (magenta) in their nucleus (yellow dotted circle) as compared to other microglia (MG) in the same region (lower panel). **(D)** Quantification of pSMAD3 intensity between CAM and MG nuclei in the same region in 4-mo-old *Pdgfb*^*ret/ret*^ mice (n = 3 mice, unpaired two-tailed *t* test, *P* = 0.0275). Age of mice: 4 mo. Scale bars: (A, C) 10 μm. Source data are available for this figure.

Gowen et al, 1999). We analyzed whether the increase in deposition of cathepsin K corresponded to a decrease in the calcification load. Isotype- or AL002a-treated littermate control *Pdgfb*^*ret/wt*^ mice did not develop vascular calcifications (Fig S2B), whereas *Pdgfb*^*ret/ret*^ mice in both groups presented with vascular calcifications (Fig 3F). We quantified the calcification load based on anti-osteopontin staining. All analyzed groups presented with high inter-individual variation as previously reported (Zarb et al, 2019). The calcification load did not differ between untreated, isotype control, or AL002a-treated groups (Fig 3G). Because calcifications, in addition to the number, grow in size with age (Keller et al, 2013, Nahar et al, 2020), we further analyzed whether the treatment affected their growth. The calcifications occurred in similar numbers across all size ranges within each group in *Pdgfb*^*ret/ret*^ mice (Fig 3H). Thus, although sustained activation of TREM2 increased cathepsin K deposition in calcifications, it had no impact on their formation or growth during the treatment period.

### Administration of AL002a does not alter mineralization of vascular calcifications

Vascular calcifications are composed of calcium phosphate that is deposited on a permissive matrix (e.g., collagen I). In addition, calcifications contain several other bone and brain proteins (e.g., osteocalcin, osteopontin, APP) (Keller et al, 2013; Zarb et al, 2019;

Nahar et al, 2020). Our analysis of calcification number and size was based on their osteopontin reactivity, but such analysis would not reveal whether administration of AL002a influenced the mineral composition of calcifications. We therefore analyzed the elemental composition of lamellar nodules in isotype and AL002a-treated *Pdgfb*^*ret/ret*^ mice using electron microscopy coupled with energy-dispersive X-ray spectroscopy (Fig 4A and B). Of note, ectopic calcifications appear as lamellar nodules under electron micros-copy (Fig 4A), as shown by others (Kodaka et al, 1994; Nahar et al, 2020; Maheshwari et al, 2022) and us (Keller et al, 2013; Maheshwari et al, 2023). These lamellar nodules, both in isotype and in AL002a-treated *Pdgfb*^*ret/ret*^ mice (Fig 4A), showed spectra with peaks that corresponded to oxygen, phosphorus, and calcium, confirming that these deposits contain calcium phosphate (Fig 4B). In addition, osteopontin-positive calcifications in isotype and AL002a-treated *Pdgfb*^*ret/ret*^ mice bound fluorescently labeled risedronate-AF647 (Fig 4C), a chemically stable inorganic pyrophosphate that binds to hydroxyapatite crystals with high affinity (Roelofs et al, 2012). Thus, administration of AL002a did not influence the mineral composition of calcifications.

### CAM retain their broad signature after AL002a antibody treatment

Previous studies have shown that TREM2 is important for survival and proliferation of microglia (Zheng et al, 2017). Interestingly,

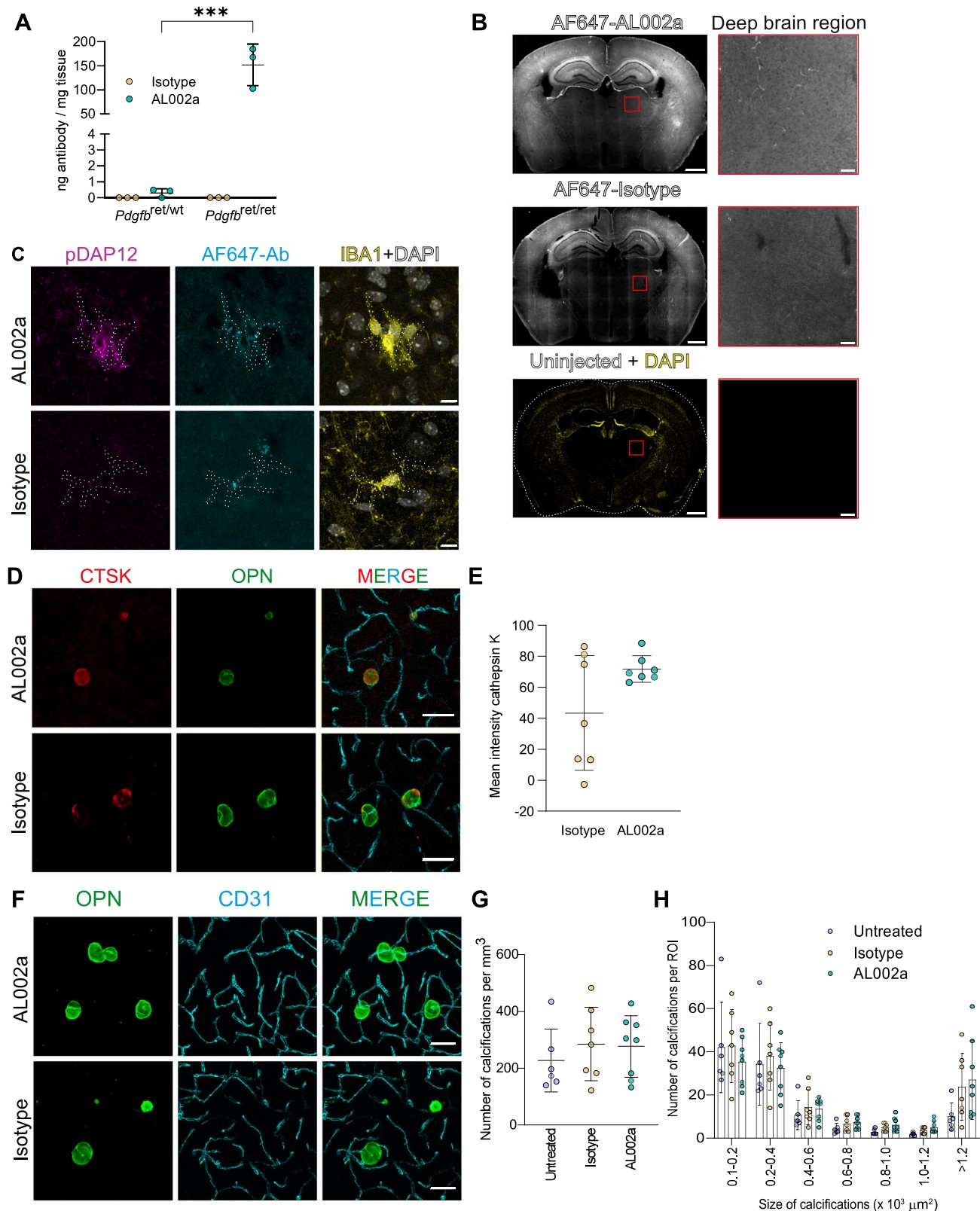

**Figure 3. TREM2-activating antibody engages with microglia and induces microglial deposition of cathepsin K in calcifications.**
**(A)** Quantification of TREM2 antibody after a single dose AL002a in the cerebrums of 2-mo-old *Pdgfb*[ret/ret] and *Pdgfb*[ret/wt] mice (n = 3). **(B)** AL002a-AF647 and isotype-AF647 are detected in 4-mo-old *Pdgfb*[ret/ret] mouse brain 1 wk post-single dose with 60 mg/kg of antibody. Zoomed-in views of the deep brain regions (red box) are shown beside. **(C)** AL002a-AF647 and isotype-AF647 (in cyan) are detected around IBA1-positive microglia in the deep brain region of 4-mo-old *Pdgfb*[ret/ret] mouse brain 1 wk post-

treatment with AL002c (mouse IgG against hTREM2) induced microglial proliferation in a preclinical mouse model of AD (Wang et al, 2020), and therefore, we asked whether the treatment with AL002a has an effect on the proliferation of microglia around calcifications in *Pdgfb*[ret/ret] mice. The number of proliferating microglia and the total number of microglia around calcifications were not altered in AL002a compared with isotype-treated *Pdgfb*[ret/ret] mice (Fig 5A). On average, 2% of microglia in the deep brain region were Ki67-positive in both isotype and AL002a-treated *Pdgfb*[ret/ret] mice (Fig 5B). There were also no differences in the total number of microglia in the deep brain region in both treatment conditions (Fig 5B). No proliferating microglia were detected in the corresponding brain region in *Pdgfb*[ret/wt] mice (Fig S2C).

We next investigated whether administration of a TREM2-activating antibody changes the expression of markers highly or specifically expressed by CAM in *Pdgfb*[ret/ret] mice. Immunohistochemical detection of CD68 (Fig 5C) and CLEC7A (Fig 5D and E) showed similar reactivity in CAM of both isotype- and AL002a-treated *Pdgfb*[ret/ret] mice, which is consistent with our previous study wherein loss of *Trem2* did not lead to the loss of CAM (Zarb et al, 2021). In addition, CAM express APOE, which did not show significantly different alteration after AL002a treatment (Fig 5D and F). CAM, unlike DAM, do not show positivity for osteopontin (Fig S1F and Zarb et al, 2021), the protein product of *Spp1* transcript, and osteopontin was also not detected in CAM after treatment with AL002a (Fig 5C). Microglia in the deep brain regions in *Pdgfb*[ret/wt] mice express CD68 in both treatment conditions (Fig S2D). Administration of the control or AL002a antibodies did not induce the expression of CLEC7A and APOE in microglia of *Pdgfb*[ret/wt] mice (Fig S2E). Thus, AL002a treatment did not alter the proliferation or expression pattern of selected CAM markers.

## Discussion

In PFBC patients, vascular calcification is a diagnostic criterion, but its role in neurodegeneration remains unclear. Our previous work has demonstrated that microglia control vascular calcification via TREM2 in a mouse model of PFBC (Zarb et al, 2021). This study investigated whether modulating microglial activity with a TREM2 agonist antibody could reduce vascular calcification in a PFBC mouse model. If successful, this would allow us to test whether reducing the calcification load could lead to an improvement of behavioral symptoms, which could open up studies on the therapeutic potential of modulating TREM2 in PFBC patients.

In this study, we extended and confirmed our previous analysis of the CAM phenotype, which was based on a limited set of selected markers (Zarb et al, 2021). snRNA-seq analysis of deep brain

microglia revealed that CAM acquire a distinct transcriptomic signature, expressing *Lpl*, *Cd74*, *Csf1*, and *Axl*, among others (Fig 1A and B). Microglia expressing the CAM signature were only found in deep brain regions where vascular calcification is present (Fig S1D and E). The CAM signature closely resembled the signature of DAM from 5X-FAD mice (Fig 1E), confirming our previous conclusion that microglia surrounding calcifications exhibit a common response similar to that observed in Alzheimer's disease and other neurodegenerative conditions (Zarb et al, 2021). Interestingly, microglia in the CAM cluster express *Spp1* (Figs 1B and S1F), also up-regulated in DAM (Keren-Shaul et al, 2017; Friedman et al, 2018; Lan et al, 2024). Although osteopontin mRNA is expressed in CAM near calcifications, the protein is not detected (Figs 5C and S1F), likely because of its rapid secretion. Notably, osteopontin protein is detectable in DAM (Qiu et al, 2023), suggesting that despite a shared transcriptional profile, microglial protein expression and signaling pathways exhibit greater complexity in various disease states. However, microglial depletion in *Pdgfb*[ret/ret] mice does not lead to the loss of osteopontin signal in calcifications (Zarb et al, 2021), indicating that even if CAM secrete osteopontin, they are not the sole source of osteopontin in calcifications.

Microglia respond to various neurological insults by shifting from their homeostatic state to an activated state (Deczkowska et al, 2018). Although some core transcriptional changes are common across these activated states, specific responses are also tailored to the nature of the insult (Candlish & Hefendehl, 2021). In 5X-FAD mice, microglia acquire a "stage I" DAM signature in a TREM2-independent manner but require a TREM2 signal to transition into "stage 2" DAM (Keren-Shaul et al, 2017). Notably, the CAM and DAM share the core transcriptomic signature (Fig 1D). Here, we show that CAM up-regulate TREM2 signaling (Fig 2A and B) as evidenced by an increase in pDAP12. Although pDAP12 generally indicates active TREM2 signaling, it is important to note that DAP12 interacts with other receptors such as SIGLEC-H (Blasius et al, 2006), which is also expressed in microglia (Konishi et al, 2017). TREM2 is essential for microglia to adopt the DAM phenotype, but its role in CAM development remains unclear. Although *Trem2* deficiency does not affect the acquisition of the CAM phenotype, defined by CLEC7A expression in *Pdgfb*[ret/ret] mice (Zarb et al, 2021), further studies, such as adult-induced microglia-specific knockouts of *Trem2*/*Dap12* in *Pdgfb*[ret/ret] mice, are needed to fully elucidate the contribution of TREM2 signaling to CAM formation. DAM and CAM differ in the activity of TGFβ signaling. Microglia suppress TGFβ signaling as they transition from a homeostatic to the DAM state in 5X-FAD mice (Krasemann et al, 2017), and DAM in human cases of AD have reduced pSMAD3 (Yin et al, 2023), indicating that DAM continue to suppress TGFβ signaling. Also, silencing TGFβ signaling by the deletion of TGFβ receptor II in microglia leads to the loss of a

---

single dose with 60 mg/kg of antibody (yellow). Microglia coincident with high AL002a-AF647 have conspicuous pDAP12 signal (magenta). **(D)** Calcifications (OPN, green) have increased deposition of cathepsin K (CTSK, red) in mice after sustained administration of AL002a (upper panel) as compared to isotype (lower panel). **(E)** Quantification of cathepsin K intensity in calcifications from AL002a-treated (n = 7) and isotype-treated (n = 7) *Pdgfb*[ret/ret] mice (unpaired two-tailed *t* test, *P* = 0.0706, n.s.). **(F)** Representative image of calcifications detected by osteopontin (green), associated with vessels detected by CD31 (cyan). **(G)** Quantification of the calcification load as measured by the number of calcification in untreated, isotype-treated, and AL002a-treated *Pdgfb*[ret/ret] mice (one-way ANOVA, F = 0.4697, *P* = 0.6326, n.s.). **(H)** Bar plots showing the frequency of calcifications per analyzed ROI classified into six size groups with surface areas between 100 and 3,000 $\mu m^2$. **(D, E, F, G, H)**: Analyzed mice (4 mo old) received eight doses of AL002a (1x per week, 60 mg/kg). Scale bars: (B) 1,000 $\mu m$; zoom in, 100 $\mu m$; (C) 10 $\mu m$; and (D, F) 50 $\mu m$. Source data are available for this figure.

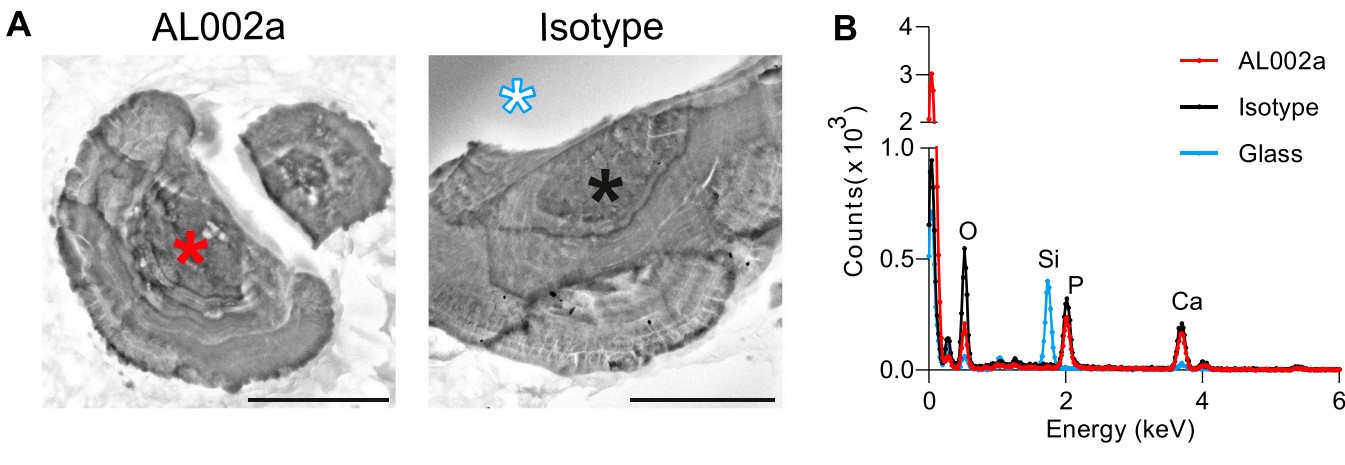

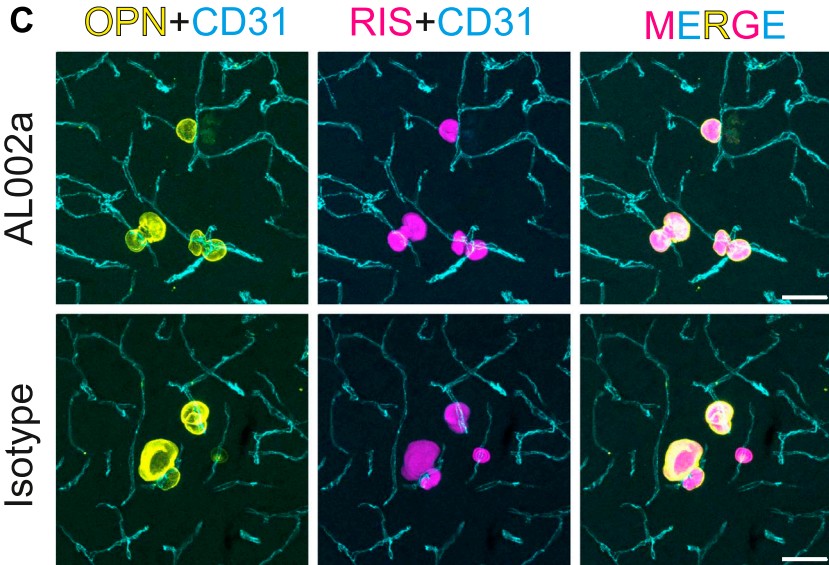

**Figure 4. Chronic treatment with AL002a does not alter the mineral composition of calcifications.**
**(A)** Scanning electron microscopy images of calcifications in AL002a- and isotype-treated *Pdgfb*^ret/ret^ mice. **(B)** X-ray dispersive emission spectrographs of calcifications (red line—AL002a-treated mouse; black line—isotype-treated mouse) show that in both treatment conditions, calcifications are composed of $Ca^{2+}$ and Pi. The peaks corresponding to each element are labeled. The blue line represents a spectrograph derived from glass (blue-white star in the isotype-treated sample), shows peaks corresponding to Si and O, and displays the specificity of the technique. **(C)** Calcifications detected by osteopontin (OPN, yellow) also stained for risedronate-AF647 (RIS, magenta), a bisphosphonate that binds to hydroxyapatite crystals found in calcifications. Scale bars: (A) 10 $\mu m$ and (C) 50 $\mu m$.
Source data are available for this figure.

homeostatic state in healthy mice (Zoller et al, 2018). CAM, on the contrary, show an increased pSMAD3 signal, as compared to microglia in their vicinity, indicative of active TGFβ signaling (Fig 2C and D). Thus, CAM differ from DAM in the maintenance of the reactive phenotype. It is plausible that CAM up-regulate TGFβ signaling specifically in response to the calcification, potentially diverging from DAM in their context-specific response. Further investigation is required to clarify the role of TGFβ signaling in CAM.

The process of how microglia remove calcifications is not known. Microglia surrounding calcifications express cathepsin K (Zarb et al, 2021), an enzyme expressed by bone-resorbing osteoclasts (Bromme et al, 1996) but absent in homeostatic microglia. Osteoclasts use cathepsin K to digest collagen and other matrix proteins in demineralized regions. Cathepsin K–deficient osteoclasts are

defective in resorbing demineralized bone (Gowen et al, 1999), and cathepsin K overexpression in osteoclasts enhances bone resorption (Kiviranta et al, 2001). It is plausible therefore that microglia secrete cathepsin K to digest the extracellular matrix present in calcifications. Consistent with our previous finding that cathepsin K expression by microglia is *Trem2*-dependent, the sustained activation of TREM2 enhanced microglial deposition of cathepsin K onto calcifications (Fig 3D). However, although cathepsin K deposition increased, this was not accompanied by the reduced number or size of calcifications in *Pdgfb*^ret/ret^ mice (Fig 3G and H). Acidification of the extracellular space at sites of bone resorption, leading to the dissolution of hydroxyapatite into its ionic constituents, is a crucial step before clearance of the demineralized matrix in bone by osteoclasts (Baron et al, 1985; Stenbeck

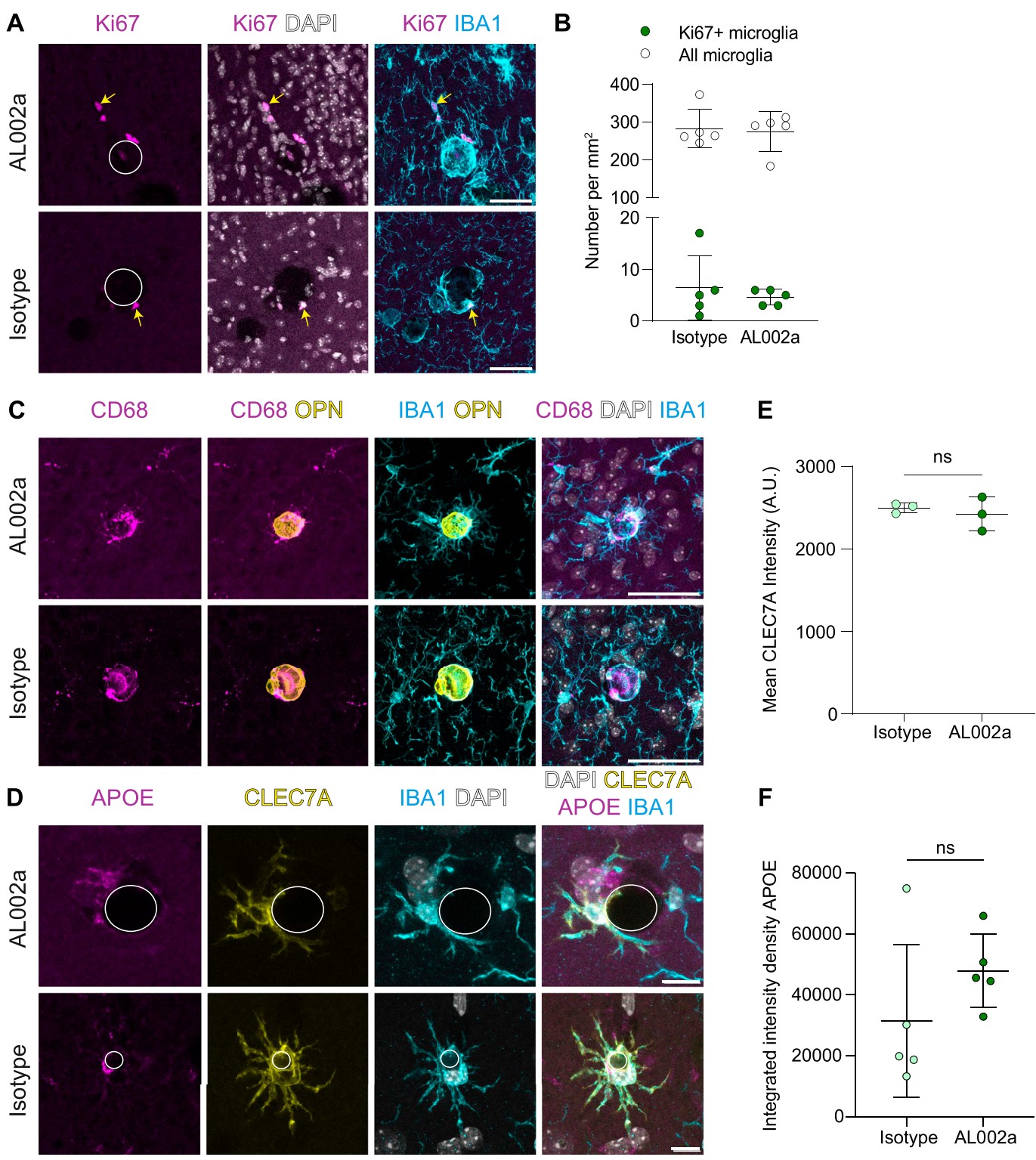

**Figure 5. Chronic treatment with AL002a does not alter microglial phenotype around calcifications.**
**(A)** CAM, positive for the proliferation marker Ki67 (magenta, yellow arrows) and IBA1 (cyan), were present around calcifications (surrounded by a white circle) after AL002a or isotype administration. **(B)** Quantification of the proliferating Ki67+ microglia (unpaired two-tailed *t* test, *P* = 0.54, n.s.) and the total number of microglia (unpaired two-tailed *t* test, *P* = 0.78, n.s.) in AL002a (n = 5)- or isotype (n = 5)-treated mice. **(C)** Expression of CD68 (magenta) in CAM (IBA1, cyan) after AL002a or isotype administration around calcifications (OPN, yellow). **(D)** Expression of CLEC7A (yellow) and APOE (magenta) around calcifications (surrounded by a white line) in AL002a- and isotype-treated mice. Scale bars: (A, C) 50 *μm* and (D) 10 *μm*. **(E)** Quantification of mean CLEC7A intensity (unpaired two-tailed *t* test, *P* = 0.58, n.s., n = 3).
**(F)** Quantification of integrated intensity density of APOE from CAM (unpaired two-tailed *t* test, *P* = 0.22, n.s., n = 5).
Source data are available for this figure.

& Horton, 2000). Whether CAM use the similar strategy to clear calcifications remains unclear and warrants investigation.

Although the full spectrum of pathways involved in preventing or removing calcifications remains unknown, TREM2 is known to regulate key microglial functions like phagocytosis (Takahashi et al, 2005; Kleinberger et al, 2014), inflammation (Jiang et al, 2014), and migration to sites of neuronal injury (Mazaheri et al, 2017). These functions could all contribute to preventing tissue calcification or controlling the growth of existing calcified foci. Existing evidence suggests that TREM2-deficient microglia are less efficient to phagocytose myelin (Cantoni et al, 2015), Aβ (Zhao et al, 2018), and hydroxyapatite crystals (Kiani Shabestari et al, 2022). In addition, activating TREM2 was beneficial in clearing Aβ plaques in the 5X-FAD mouse model of AD (Price et al, 2020) and myelin debris in a cuprizone-induced model of demyelination injury (Cignarella et al, 2020). However, activating TREM2 signaling did not lead to clearance of calcifications (Fig 3F–H). TREM2 has also been shown to be crucial for microglial proliferation and survival (Zheng et al, 2017), and activating TREM2 using AL002c, mouse IgG against human TREM2, induced proliferation of specific microglial populations in a preclinical mouse model of AD (Wang et al, 2020). Another anti-TREM2–activating antibody induced a transient, dose-dependent increase in microglial proliferation both in vivo and in vitro (van Lengerich et al, 2023). The increase in proliferation was observed 48 h after antibody administration but not detected 1 wk after. We observed that treatment with AL002a did not enhance the number of microglia positive for Ki67 or the total number of microglia in calcified regions (Fig 5A and B). However, our analysis was conducted 1 wk after the last dose, and therefore, we cannot rule out transient changes in microglial proliferation. We observed an increase in APOE signal in CAM after anti-TREM2 treatment, although this increase was not statistically significant. This is consistent with other studies that suggest the *Trem2*-mediated induction of disease-associated profiles is associated with increased *Apoe* (Krasemann et al, 2017; Sala Frigerio et al, 2019). The role of TREM2 signaling in CAM remains to be fully understood. It is possible that additional activation of TREM2 signaling has limited effects in CAM under unimpaired TREM2 function because CAM already have activated TREM2 signaling (Fig 2A and B). Further studies should investigate whether TREM2 activation can mitigate the calcification pathology in *Pdgfb*<sup>ret/ret</sup> mice with *Trem2* haploinsufficiency (Zarb et al, 2021) or in older mice with reduced microglial activity or function (Li et al, 2023).

Although the mechanism of microglial removal of calcifications remains unclear, several lines of evidence from studies of human genetic diseases and preclinical studies suggest the possibility of microglial clearance of calcifications. Notably, longitudinal imaging studies showed that infants infected prenatally with the Zika virus developed brain calcifications that resolved with time (Petribu et al, 2017). Although the role of microglia in removing these calcifications remains unclear, their clearance coincided with increased microglial phagocytosis of synapses in infancy (Menassa et al, 2022). Recent studies have shown that mice lacking microglia (*Cs1r*<sup>ΔFIRE</sup> mice) develop vascular and parenchymal calcification, which can be reversed by transplanting human iPSC-derived microglia in adulthood or prevented by transplanting murine microglia in early postnatal stages (Chadarevian et al, 2024; Munro et al, 2024). The

microglia-deficient *Cs1r*<sup>ΔFIRE</sup> mouse model, coupled with microglial engraftment, offers a powerful tool for unraveling the mechanisms by which microglia contribute to calcification resolution.

Building on previous findings that functional TREM2 is necessary for microglial regulation of the calcification load in the PFBC *Pdgfb*<sup>ret/ret</sup> model (Zarb et al, 2021), our current study suggests that mere activation of the TREM2 pathway without considering other factors is insufficient to reduce vascular calcification. Further research on microglial control of ectopic calcification is needed to identify potential pathways for halting or reducing the vascular or parenchymal calcification in PFBC and other brain diseases.

# Materials and Methods

## Mice

In this study, 2- to 5-mo-old mice of either sex were used, and similar findings were reported for both sexes. The B6.*Pdgfb*<sup>tm3Cbet</sup> (*Pdgfb*<sup>ret/ret</sup>—mutant; and *Pdgfb*<sup>ret/wt</sup>—control) mouse strain (Lindblom et al, 2003) was used. Mice were housed in individually ventilated type 2L cages under optimized hygienic conditions. The cages were maintained in a temperature- and humidity-regulated environment, enriched with standard bedding material, red house, tissues, and paper crinkles, and housed in a room under 12/12-h light/dark cycle. This study was conducted according to the standard operating procedures approved by the Cantonal Veterinary Office Zurich (license ZH194/2020).

## Administration of AL002a and isotype control

2-mo-old mice were injected intraperitoneally with 60 mg/kg AL002a in 20 mM His, 7.5% (wt/vol) sucrose, and 0.02% Tween-80, pH 6; or isotype control (IgG1) in 20 mM His, 7.5% (wt/vol) sucrose, and 0.02% Tween-80, pH 6. AL002a and isotype control (Cignarella et al, 2020; Price et al, 2020) were provided by Alector. Both antibodies were on a murine backbone and contained an Fc domain. For acute studies, mice were euthanized 1 wk after the administration of a single dose of either antibody. For chronic administration, mice were injected once a week beginning at the age of 2 mo for 8 wk. 1 wk after the final dose, the mice were deeply anesthetized with an overdose of xylazine/ketamine in saline and perfused transcardially with cold PBS (pH 7.4). The olfactory bulbs and forebrain regions consisting of the rostral cerebrum were frozen in dry ice and stored at –80°C. The caudal cerebrum and cerebellum were fixed in 4% PFA in PBS (pH 7.4) for 6 h at 4°C. After fixation, the tissue was washed with PBS (pH 7.4) and stored in PBS containing 0.01% (wt/vol) sodium azide at 4°C until further use.

## Single-nucleus RNA sequencing and analysis

### Nucleus isolation from the frozen mouse brain tissue
Six female *Pdgfb*<sup>ret/ret</sup> mice and six *Pdgfb*<sup>ret/wt</sup> mice (five females, one male) aged 5 mo were used for single-nucleus RNA-sequencing analysis. Mice were deeply anesthetized with an overdose of xylazine/ketamine in saline and perfused with cold PBS containing

10 U/ml of heparin. The brains were macrodissected on ice into cortical (non-calcification–prone) and deep brain (calcification-prone) regions and frozen on dry ice. Tissue was processed for single-nucleus isolation as previously described in Zhou et al (2020). Isolated mouse nuclei were subjected to droplet-based 5′ end massively parallel single-cell RNA sequencing using Chromium Single Cell 5′ Reagent Kits per the manufacturer's instructions (10x Genomics). The dually indexed libraries were sequenced using the Illumina NovaSeq 6000 sequencer at the McDonnell Genome Institute, USA. Sample demultiplexing, barcode processing, and single-nucleus calling were performed using the CellRanger analysis pipeline (v. 5.0.1, 10x Genomics). CellRanger count was used with the –include-introns parameter to align sequencing reads of each sample to the mm10 reference genome (version refdata-gex-mm10-2020-A downloaded from the 10x Genomics website), quantify reads, and filter reads with a quality score below 30. The count matrices generated by CellRanger were imported into R (version 4.1.0), and quality check was performed on the count matrix output from CellRanger using a homemade function. Briefly, nuclei with detected features above 500 and below 7,000 and mitochondrial gene expression less than 20% were retained. Using the Seurat package for R version 4.0 (Hao et al, 2021), a Seurat object was created from the filtered count matrix and the data were log-normalized with a scale factor of 1,000. The scaled expression values of the top 2,000 variable features were used for principal component analysis. UMAPs were plotted using 20 principal components (for cluster analysis of all the deep brain nuclei) or 30 principal components (for microglia). These same numbers of input principal components were used for cell clustering. The Find-Neighbors function in Seurat was used to create the k-nearest neighbor graph. This was followed by the FindClusters function with the resolution parameter set to 0.01 for all deep brain nuclei. The low-resolution parameter aided in obtaining broad communities of nuclei (neurons, astrocytes, microglia, etc.). After removing nuclei, the FindAllMarkers function was used to determine the differential gene expression between each cluster against all other clusters. The Wilcoxon rank sum test was used to determine statistical significance. For the CAM cluster, differential gene expression was calculated by comparing with the two homeostatic microglial clusters. The $P$-value was adjusted using the Bonferroni correction. Volcano plots were generated using the EnhancedVolcano package in R (Blighe et al, 2024). Genes with an average $\log_2$ fold change greater than 1.0 and adjusted $P$-value less than 0.05 were used for gene enrichment analysis.

### Comparison of the transcriptomes of CAM and DAM

To obtain the gene signatures of DAM, the publicly available 7-mo dataset from Zhou et al (2020) was used. This dataset was chosen because of the closeness in age to the mice used in the present single-nucleus RNA-sequencing study. Briefly, the WT and 5X-FAD single-nucleus sequencing data were reanalyzed with the same parameters used in this study. The microglial cluster was identified as described above and used for further analysis. The Sridhar (current study) and Zhou microglial datasets were analyzed with the alignment pipeline described in Seurat v3. Briefly, anchor points or features common to both datasets were identified followed by integration of the two datasets. The thus

generated data were then scaled, and dimensions were reduced with PCA. A UMAP was created on the first 30 principal components. Integrated clusters were identified using the FindClusters function using a resolution parameter of 0.4. FindAllMarkers was used to identify the top features that distinguished the individual clusters using default parameters. Cluster-defining genes were plotted across the different clusters using the DotPlot function.

### Gene set enrichment analysis with g:Profiler2

The web-based gene set enrichment analysis tool g:Profiler2 (Reimand et al, 2007) was used to detect overlap between differentially expressed genes and Gene Ontology: Biological Process and Cellular Component, KEGG pathways, REACTOME, and Wiki pathway databases.

### AL002a quantification and distribution in brain

The amount of anti-TREM2 antibody in the brain was quantified as described previously (Cignarella et al, 2020). To assess bioavailability and distribution of the injected antibodies, both AL002a and isotype were fluorescently labeled with Alexa Fluor 647 (AF647). First, both antibodies were subjected to buffer exchange with PBS using Zeba spin desalting columns (89889; Thermo Fisher Scientific). Then, AL002a and isotype were labeled with AF647 using a protein labeling kit (A20173; Thermo Fisher Scientific). The degree of labeling calculated as the ratio of the quantity of dye to protein (i.e., moles of dye/moles of protein) was estimated to be 3 for AL002a and isotype as is standard from the kit. Two male $Pdgfb^{ret/ret}$ mice were injected with 60 mg/kg of labeled AL002a-AF647 or isotype-AF647, respectively. 1 wk post-injection, mice were perfused as described above. The brains were fixed in 4% PFA as described above and later sectioned and analyzed by immunohistochemistry and fluorescence microscopy.

### Immunohistochemistry

Fixed mouse brains were cut using the Leica VT1000S vibratome into sections of 60 $\mu$m thickness. Free-floating sections were used for antibody staining. Briefly, the sections were blocked and permeabilized in the blocking/permeabilizing buffer (PS) (1% BSA and 2% vol/vol Triton X-100 in PBS, pH 7.4) for 24 h at 4°C. The primary antibody was diluted in PS2 (1:1 dilution of the blocking/permeabilizing buffer in PBS). Each section was incubated in a 24-well plate with 150 $\mu$l of antibody cocktail for 3 d at 4°C. After the incubation period was completed, the tissue sections were washed three times with PS-2 for 10 min before the secondary antibody cocktails (raised in donkey, minimum cross-reactivity, 1:600; Jackson ImmunoResearch) were added. The sections were incubated in the secondary antibody cocktail overnight followed by washing twice with PS2. The sections were then stained with 4′,6-diamidino-2-phenylindole dihydrochloride (DAPI, D9542, stock concentration 10 mg/ml, diluted 1:10,000 in PBS; Sigma-Aldrich) for 10 min. After a final wash with PBS, the sections were mounted onto glass slides in ProLong Gold Antifade Reagent (P36930; Thermo Fisher Scientific). The antibodies used in this study are listed in Table S2.

## Fluorescence in situ hybridization and immunohistochemistry

A set of ssDNA probes recognizing the coding sequence of *Spp1* compatible with the MUSE amplification technology (arcoris bio AG; Table S1) was used. Sixty-micrometer-thick brain slices were sectioned using a vibratome in cold PBS in a vibratome chamber maintained on ice. Sections were incubated in 200 μl blocking buffer overnight composed of RNase inhibitor (3335399001, 0.2 U/ml; Merck), 1% RNase-free BSA (Cat No. 3737.3; Carl Roth), and 2% Tween-20 in sterile PBS. Next, the primary antibodies were diluted in 400 μl of blocking buffer and incubated over two nights. After incubation, the sections were washed in 500 μl PBS three times for 10 min. The bound primary antibody was fixed in 4% PFA for 10 min at room temperature. After washing off the fixative with PBS, the sections were washed three times for 10 min with 300 μl SSCT buffer containing 2X SSC buffer diluted from a 20X SSC stock (15557044; Thermo Fisher Scientific) in RNase-free water and 0.1% Tween-20. Sections were then incubated in 600 μl prewarmed hybridization buffer (arcoris bio AG, Hyb reagent) at 37°C for 1 h. Sections were then incubated with the *Spp1* probes diluted 100X in 300 μl Hyb reagent and incubated overnight at 37°C. All incubation steps at 37°C were carried out in a humidified chamber. The amplification steps were performed using MUSE chemistry in a two-step process per the standardized protocols provided by the manufacturer arcoris bio AG. Each amplification step was performed at room temperature for 4 h. Sections were incubated with appropriate secondary antibodies as described above and finally counterstained with DAPI, washed with PBS, and mounted in ProLong Gold Antifade Reagent (P36930; Thermo Fisher Scientific). Images were acquired using a Leica SP8 inverse confocal microscope (63x oil objective, NA 1.41).

## Image acquisition and analysis

Images of immunohistochemical stains were acquired with a confocal microscope (Leica SP8: 20x air objective, numerical aperture [NA] 0.7; and 63x oil objective, NA 1.4) or slide scanner (Zeiss Axio Scan.Z1: 10x air objective; NA 0.45). Images were analyzed using image processing software Imaris 9.2.0. (Bitplane) and Fiji (ImageJ 1.54f) software.

### Quantification of the calcification load

The calcification load was quantified on 10 coronal vibratome sections spanning the regions with the least variability in the calcification load (Zarb et al, 2019). The sections were stained with antibodies against osteopontin to locate calcifications and CD31 to ensure the osteopontin signal was vessel-associated. Sections were imaged using Zeiss Axio Scan.Z1 with a 10x objective, NA 0.45. The images were further analyzed with Bitplane Imaris 9.2.0. Each image was also carefully checked, and spurious spots (non-specific secondary aggregates on tissue or non-calcification–related osteopontin expression) were removed. Each calcification in a standard region of interest (ROI) was identified using the Surfaces function in Imaris. The statistics function was used to obtain a count of the total number of surfaces (i.e., the number of calcifications) and to generate a csv file containing the area of each individual surface (referred to as the size of the calcification). These

values were normalized to the volume of the analyzed ROI and averaged by the number of analyzed ROIs per individual mouse.

$$\text{Number of calcifications per mm3} = \frac{\Sigma \text{Calicifications per ROI}}{\text{Volume of ROI*number of analyzed ROIs}}.$$

$$\text{Volume of ROI} = X \text{ dimension of ROI in } \mu m *$$
$$Y \text{ dimension of ROI in } \mu m * T \text{hickness of tissue section}.$$

To obtain a distribution of the calcification sizes, the surface area of individual nodules was classified into six size bins: 100–600, 600–1,200, 1,200–1,800, 1,800–2,400, 2,400–3,000, and >3,000 $\mu m^2$ and the number of calcifications in each bin was calculated. The number was then averaged to the number of ROIs analyzed for each mouse.

$$\text{Number of calcifications per ROI for a given size bin} = \frac{\Sigma \text{Calicifications per ROI}}{\text{Number of analyzed ROIs}}.$$

### Quantification of signal intensity of pDAP12, pSMAD3, cathepsin K, and APOE staining

Confocal laser scanning microscopy images were acquired using the Leica SP8 microscope, and the images were quantified with Fiji (ImageJ 1.54f). For pDAP12 and pSMAD3 quantifications, CAM were identified as CLEC7A-high microglia and microglia in the vicinity that had low to no expression of CLEC7A were imaged for controls. For pDAP12 and APOE quantification, the IBA1 signal was used to create a mask and applied on the pDAP12 or APOE channel to obtain microglial pDAP12 or APOE signal and eliminate background staining. APOE and pDAP12 expression was quantified from five and three *Pdgfb*[ret/ret] mice, respectively. For pSMAD3 quantification, an ROI was drawn around the DAPI signal of the microglia analyzed and the pSMAD3 signal was estimated (data from three *Pdgfb*[ret/ret] mice). The corresponding background signal, signal found in regions without nuclei, was subtracted from this value for every microglia. For cathepsin K quantification, ROIs were drawn around osteopontin calcifications and the signal intensity was measured. Background intensity in regions without calcification was subtracted from the intensity for each image.

### Quantification of proliferating microglia and CLEC7A expression

Images were acquired using Axio Scan.Z1, 20X objective, NA 0.8, and the cell detection function in QuPath (v. 0.3.2) was used to identify microglia in raw images (.czi files). Microglia were identified based on the expression of IBA1. The following parameters were applied to detect microglia: requested pixel size (microns)—1.0; background radius (microns)—3.0; median filter radius—3.0; sigma—0.5; minimum area—10.0; and maximum area—400.0. The intensity of the IBA1 signal higher than 200 was identified as microglia. Then, the cellular Ki67 intensity above 5,000 was used as a parameter to identify proliferating microglia. The number of positive (Ki67+ microglia) and total (all microglia) detections was normalized to the area analyzed. Data from two sections per mouse were averaged, and data from 5 (three males and two females) mice per treatment

condition were analyzed. For CLEC7A expression analysis, microglia were identified as described above (the IBA1 signal threshold was set at 500) and cells with mean cellular CLEC7A intensity above 2000 were identified as CAM. The mean cytoplasmic CLEC7A intensity from the identified CAM was calculated for each mouse and analyzed from three mice per treatment condition.

### Energy-dispersive X-ray spectroscopy

SEM analysis was carried out on paraffin wax samples. The wax was removed using pure xylene for two 10-min intervals. The slides were then mounted on sample holders using a double-sided carbon adhesive tape, painted with silver conductive paint, and coated with a 5-nm carbon layer. A Carl Zeiss LEO 1530 was used at accelerating voltages of 10 kV for SEM imaging, which included secondary electron and backscattering electron modes. Energy-dispersive X-ray spectroscopy (EDX) analysis was carried out using Oxford Instruments EDX.

### Statistical analysis

Quantified values are represented as means ± SD. The following statistical tests were performed with Prism 9 software (GraphPad). The following statistical analyses were used: $t$ test (unpaired and two-tailed) and one-way analysis of variance (ANOVA) with Dunnett's multiple comparisons test. $P$-values < 0.05 were considered significant.

# Data Availability

The snRNA-seq data consisting of fastq raw data files and outputs from CellRanger v. 5.0.1. pipeline have been deposited in the Gene Expression Omnibus (GEO) database with the accession number GSE263392. Gene expression lists and supporting data values have been provided within the article. Additional data related to this article may be requested from the authors.

# Supplementary Information

# Acknowledgements

This study was supported by the grants to A Keller from the Swiss National Science Foundation (310030_188952, 320030_228518), Dementia Research Switzerland—Synapsis Foundation and Choupette Foundation (2019-PI02), ERA-NET NEURON (32NE30_213467), and Novartis FreeNovation; and to S Sridhar from the GRC Career Grant, University of Zurich. We thank Irina Abakumova and Marianne König from the Institute of Neuropathology, University Hospital Zurich, for technical support. We thank the Center for Microscopy and Image Analysis, University of Zurich, for providing access to microscopes and image analysis software. Sequencing was performed at the Elizabeth H and James S. McDonnell III Genome Institute at Washington University.

## Author Contributions

S Sridhar: conceptualization, data curation, formal analysis, investigation, visualization, methodology, and writing—original draft, review, and editing.
Y Zhou: data curation, formal analysis, investigation, and methodology.
A Ibrahim: data curation, formal analysis, investigation, and methodology.
S Bertazzo: data curation, formal analysis, investigation, and methodology.
T Wyss: data curation, software, formal analysis, and investigation.
A Swain: data curation, investigation, and methodology.
U Maheshwari: data curation, formal analysis, investigation, and methodology.
S-F Huang: data curation, formal analysis, investigation, and methodology.
M Colonna: resources and supervision.
A Keller: conceptualization, resources, formal analysis, supervision, funding acquisition, investigation, methodology, project administration, and writing—original draft, review, and editing.

## Conflict of Interest Statement

M Colonna is a member of the Vigil Scientific Advisory Board and NGM Bio, is a consultant for Cell Signaling Technology, and has received research grants from Alector, Amgen, Pfizer, Vigil, NGM Bio, and Ono for activities not related to the findings described in this publication. In addition, M Colonna has a pending patent related to TREM2. A Ibrahim is an employee of Alector, USA. All other authors declare that they have no competing interests.

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
