## [Reviewer comments · Life Science Alliance]

Life Science Alliance

Targeting TREM2 signaling shows limited impact on cerebrovascular calcification

Sucheta Sridhar, Yingyue Zhou, Adiljan Ibrahim, Sergio Bertazzo, Tania Wyss, Amanda Swain, Upasana Maheshwari, Sheng-Fu Huang, Marco Colonna, and Annika Keller

DOI: <https://doi.org/10.26508/lsa.202402796>

Corresponding author(s): Annika Keller, University of Zurich

Review Timeline:

Submission Date:	2024-04-26
Editorial Decision:	2024-07-24
Revision Received:	2024-10-01
Editorial Decision:	2024-10-10
Revision Received:	2024-10-15
Accepted:	2024-10-16

Transaction Report:

July 24, 2024

Re: Life Science Alliance manuscript #LSA-2024-02796

Dr. Annika Keller
University Hospital of Zurich
Division of Neurosurgery, Zürich University
Wagistrasse 12
Zürich CH-8952
Switzerland

Dear Dr. Keller,

Thank you for submitting your manuscript entitled "Targeting TREM2 signaling shows limited impact on cerebrovascular calcification" to Life Science Alliance. The manuscript was assessed by an expert reviewer, whose comments are appended to this letter. We invite you to submit a revised manuscript addressing the Reviewer comments.

When submitting the revision, please include a letter addressing the reviewer's comments point by point.

Thank you for this interesting contribution to Life Science Alliance. We are looking forward to receiving your revised manuscript.

Sincerely,

B. MANUSCRIPT ORGANIZATION AND FORMATTING:

Reviewer #1 (Comments to the Authors (Required)):

In their manuscript entitled „Targeting TREM2 signaling shows limited impact on cerebrovascular calcification" Sridhar et al address the question whether activating TREM2 signaling by a monoclonal anti-TREM2 antibody can rescue brain vessel calcification in *Pdgfb(ret/ret)* mice. This report is a logical follow up to the previous publication from the same laboratory (Zarb et al, 2021) where the authors showed that brain vessel calcification in *Pdgfb(ret/ret)* mice is increased upon TREM2 loss-of-function. Overall, the manuscript provides a clear rationale and is well structured. Nevertheless, the manuscript can be significantly improved in order to enable readers to better judge the relevance of the reported results. This reviewer, therefore, recommends publication once the following questions and points have been appropriately addressed.

Major points:

1. The authors report that TREM2 activation has limited impact on cerebrovascular calcification in *Pdgfb(ret/ret)* mice. This reviewer wonders whether the outcome may depend on calcification load at the time the antibody treatment is started. It would therefore be helpful to remind the non-expert readership of some basic features of this model, e.g., at what mouse age does deposition of calcium-containing deposits start and at what age does it reach a plateau? Consequently, the authors should discuss whether treatment of this mouse line at a different age may lead to a more favorable (or even worse) outcome.
2. Similar to the first point this reviewer wonders whether the authors conducted snRNA-seq analyses at different mouse ages. Is the proportion of CAM increasing as the mice are ageing? In addition, is it possible that at the time antibody treatment is initiated (i.e. when mice are 2 months old) microglia activation is already plateauing such that further activation by administration of an anti-TREM2 antibody is no longer possible?
3. In the Materials and Methods section the authors should state whether the antibodies they administered were effectorless (i.e. Fc null) or not? If not, they authors should carefully discuss if and to what extent this may affect the outcome of the study. The authors should also mention whether the antibodies were used on a mouse backbone (which this reviewer thinks should be the case).
4. The authors should define in more detail what they mean when they talk about „deep brain" throughout the manuscript. Does this contain thalamus and midbrain (compare to previous publication from the same group: Zarb et al, 2021)?
5. Related to Figure 1: it is surprising that TREM2 and TYROBP (i.e. DAP12) are not among the differentially expressed genes (DEGs) when comparing CAM and homeostatic microglia (see list of DEGs in supplementary Excel file). This could raise questions as to the relevance of TREM2 signaling for CAM function. This should be addressed by the authors. This reviewer agrees that an increase in pDAP12 signal (Figure 2A) is generally supportive of increased TREM2 signaling in CAM. However, the authors should briefly point out in the Discussion that DAP12 is a co-receptor of several receptors including TREM2 and that independent confirmation of increased TREM2 signaling in CAM is desirable.
6. In regard to Figure 1D, the authors mention *Fmn1* explicitly in the text. However, *Fmn1* cannot be found in Figure 1D. Please clarify.
As a more general comment to Figure 1D: several of the annotations are unreadable. Therefore, the Figure should be improved such that (i) annotations are readable and (ii) it is clear which data point is annotated.
7. In Figure 1F, the authors annotate an immature microglia population („im_MG"). However, in the main text of the manuscript, this population is not discussed at all. Please clarify.
8. In Figure 2A, the authors stain for CLEC7A as a marker for CAM. Surprisingly, the corresponding gene is not among the DEGs when comparing CAM and homeostatic microglia (see list of DEGs in supplementary Excel file). Nevertheless, *Clec7a* was among the DEGs in an RNA-seq analysis in the previous publication from the same group: Zarb et al, 2021. This should be explained.

9. This reviewer agrees that the very high anti-TREM2 antibody amounts as reported in Figure 3A could be due to increased BBB permeability in the *Pdgfb(ret/ret)* mouse model. This would result in much higher antibody concentrations in the brain than would normally be possible if the BBB was intact. Is it therefore possible that CAM are exhausted after 8 weeks of treatment simply because they are being overactivated? This should be discussed. Performing snRNA-seq analysis upon chronic AL002a treatment could be quite informative.

10. On top of page 9, the authors mention that isotype-A647 colocalized with IBA1 positive microglia. Why does the isotype antibody colocalize with microglia? This is difficult to understand and should be explained.

11. In Figure 5D, this reviewer finds APOE stainings not convincing enough to conclude that APOE level is not altered upon AL002a treatment. Do the authors have quantifications to support their conclusion?

12. Which brain region was analyzed in Figure 3A? Is this whole brain? This needs to be mentioned.

13. Are the images shown in Figure 3C taken from the deep brain region? Please clarify.

Minor points:

1. In the beginning of the Introduction, the authors talk about „specific conditions" and „pediatric disorders" (toward end of page 2). This needs to be more concrete.

2. On page 6, it should be „Selplg" instead of „Selpg".

3. On page 6, what do the authors mean by „genome transcriptomes"?

4. On page 7, what is the difference between „microglia" and „non-CAMs"? The authors should be consistent in regard to nomenclature.

5. On page 8, the authors mention two times that the isotype antibody was not detected in the immunoassay which detects the anti-TREM2 antibody (i.e. AL002a). Mentioning it once is enough.

6. In Figures 2 and 5, it would be desirable to show separate IBA1/OPN stainings.

7. Please add mouse age and mouse N to Figure 1 legend.

8. Please add mouse age to Figure 2 legend.

9. Figure 3 would benefit from a panel showing a scheme of the chronic treatment paradigm.

10. Please add mouse age to legend of Figures 3A-C.

11. In Figure 3G, the error bars are most likely SEM and not SD (in the Materials and Methods the authors state that all Figures show SD).

12. In Figure 3, it seems that data as shown in panels A-C are derived from a single antibody dose study while the data in panels D-H are derived from the multiple dose study. This should be clarified in the figure legend.

13. In Figure 4B, it seems that oxygen (O) content in calcifications in AL002a treated mice is lower compared to isotype treated mice. Please comment.

14. In the legend of Figure 5B one can find only one t-test-associated p value. Is it the same for both comparisons („Ki67+ microglia" and „all microglia") or is one p value missing? Please clarify.

We thank you for the opportunity to review our manuscript Sridhar et al., “*Targeting TREM2 signaling shows limited impact on cerebrovascular calcification*”.

Summary of changes, modifications in the manuscript:

1. We have modified text (modified text is in red) and provided clarifications. We hope that these changes have improved the manuscript's readability.
2. We have quantified CLEC7A and APOE (reviewer's major point 11) expression in control IgG and AL002a treated mice. This data has been added to the manuscript (Fig. 5E, F).
3. We have performed validation of snRNAseq data by performing *in situ* hybridization, showing that calcification-associated microglia (CAMs) express *Spp1*, in combination with immunohistochemical detection of microglia (IBA1) and calcification (OPN) (Supplementary Figure S1F). While the reviewer did not explicitly request confirmation of this gene's expression, they raised several questions about the snRNA seq data that could be addressed without additional experimentation. Our finding that *Spp1* is expressed by CAMs is important information and shows that while osteopontin mRNA can identify CAMs, osteopontin protein detection cannot be used. As discussed in the Discussion section, microglial protein expression and signaling pathways exhibit greater complexity in disease states, despite a shared transcriptional profile. This conclusion is further supported by our observed differences in activity of TGF- β signaling which in calcification-associated microglia but downregulated in DAMs in AD mouse models.
4. In our updated manuscript we have cited to two recent studies that provide compelling evidence for the protective effects of microglia against cerebral calcification in *Csfr1^{ΔFIRE}* mice, that lack microglia and develop cerebral calcifications, and where the transplantation/presence of microglia prevents the occurrence of calcification pathology. These studies further support the role of microglia in regulating brain calcification, warranting their citation (Introduction: page 3 lines 71-73, Discussion – see page 16 line 421 – 426).

Other experiments performed during revision, presented below but not included in manuscript:

1. We have performed bulk RNA seq analysis of isolated microglia in untreated, IgG and AL002a treated mice in response to reviewer's major point 9, *Rebuttal Figure 1 and 2*.

Rebuttal letter

We thank the reviewer for the careful reading of the manuscript and constructive criticism. We have modified the manuscript accordingly and we hope we have addressed the raised points appropriately. The reviewer text is in black, our responses are in blue.

Point by point response to the reviewer's comments:

In their manuscript entitled „Targeting TREM2 signaling shows limited impact on cerebrovascular calcification" Sridhar et al address the question whether activating TREM2 signaling by a monoclonal anti-TREM2 antibody can rescue brain vessel calcification in *Pdgfb(ret/ret)* mice. This report is a logical follow up to the previous publication from the

same laboratory (Zarb et al, 2021) where the authors showed that brain vessel calcification in *Pdgfb(ret/ret)* mice is increased upon TREM2 loss-of-function. Overall, the manuscript provides a clear rationale and is well structured.

-We are happy that the reviewer finds the manuscript well structured and the rationale for the study clear.

Nevertheless, the manuscript can be significantly improved in order to enable readers to better judge the relevance of the reported results. This reviewer, therefore, recommends publication once the following questions and points have been appropriately addressed.

- We have addressed raised questions and the modified text is in red.

Major points

1. The authors report that TREM2 activation has limited impact on cerebrovascular calcification in *Pdgfb(ret/ret)* mice. This reviewer wonders whether the outcome may depend on calcification load at the time the antibody treatment is started. It would therefore be helpful to remind the non-expert readership of some basic features of this model, e.g., at what mouse age does deposition of calcium-containing deposits start and at what age does it reach a plateau? Consequently, the authors should discuss whether treatment of this mouse line at a different age may lead to a more favorable (or even worse) outcome.

- The reviewer raises a valid point. Longitudinal studies where anti-TREM2 antibody administration is initiated at different ages (e.g., shortly after weaning, when the calcification load is smaller) would provide additional insights into the outcome. In an ideal scenario, we would have preferred to conduct the study using longitudinal *in vivo* monitoring of vascular calcification in each mouse with a suitable imaging modality. However, this approach was not feasible. Therefore, for our study, we selected a starting age of 2 months and an end point of 4 months because we had baseline values for this PFBC model. This has now been noted in the manuscript. In addition, we have added a note regarding the age-dependent growth of calcifications (page 9, lines 235-239). Since the timing of the calcification plateau remains uncertain - it appears to continue growing in all PFBC models up to 16 months, the last time point studied - we have not commented on it.

2. Similar to the first point this reviewer wonders whether the authors conducted snRNA-seq analyses at different mouse ages. Is the proportion of CAM increasing as the mice are ageing? In addition, is it possible that at the time antibody treatment is initiated (i.e. when mice are 2 months old) microglia activation is already plateauing such that further activation by administration of an anti-TREM2 antibody is no longer possible?

- snRNA-seq data is available only for 5 months old mice. It is likely that the number of CAMs increases as the number and size of calcifications grow, as calcifications are invariably surrounded by CAMs. Unfortunately, a reliable quantitative method for estimating the proportion of CAMs in each mouse is lacking, as the individual calcification load is variable and increases with age, and individual CAMs are difficult to isolate for cell number quantification (e.g., using flow cytometry).

Concerning the question whether microglia activity is plateauing at the age of 2 months in this PFBC mouse model: Microglia continuously proliferate around calcifications (Figure 3D-E in Zarb et al., 2021 (PMID: 33637522), and this manuscript Figure 5A). Whether new CAMs are derived from existing CAMs, from surrounding microglia, or from both is currently unresolved. However, given the observation that new calcifications in new brain regions are

always surrounded by microglia with CAM profile as assessed by immunohistochemistry (e.g., positive for CLEC7A, reduced expression of P2RY12, upregulation of IBA1 etc.), it is likely that they originate from homeostatic microglia nearby. This suggests that the microglial activation potential remains high in this mouse model.

Regarding the further activation of microglia via TREM2. Since the TREM2 signaling pathway is already active in CAMs in this mouse model, further activation may not be possible in mice aged 2-4 months, as discussed on page 15, lines 409-411. Whether TREM2 activity in CAMs decreases with age needs to be investigated.

3. In the Materials and Methods section the authors should state whether the antibodies they administered were effectorless (i.e. Fc null) or not? If not, they authors should carefully discuss if and to what extent this may affect the outcome of the study. The authors should also mention whether the antibodies were used on a mouse backbone (which this reviewer thinks should be the case).

- The administered antibodies were bivalent monoclonal antibodies on a mouse backbone with a functional Fc domain. This information has been added to Materials and Method section (Administration of AL002a and isotype control). It has been suggested that murine IgG has limited effector functions (PMID: 26497511). In the current study, neither the use of AL002a or isotype IgG lead to changes in the calcification load when compared with untreated controls.

4. The authors should define in more detail what they mean when they talk about „deep brain" throughout the manuscript. Does this contain thalamus and midbrain (compare to previous publication from the same group: Zarb et al, 2021)?

-This is now clarified in the legend for Supplementary Figure 1A.

5. Related to Figure 1: it is surprising that TREM2 and TYROPB (i.e. DAP12) are not among the differentially expressed genes (DEGs) when comparing CAM and homeostatic microglia (see list of DEGs in supplementary Excel file). This could raise questions as to the relevance of TREM2 signaling for CAM function. This should be addressed by the authors. This reviewer agrees that an increase in pDAP12 signal (Figure 2A) is generally supportive of increased TREM2 signaling in CAM. However, the authors should briefly point out in the Discussion that DAP12 is a co-receptor of several receptors including TREM2 and that independent confirmation of increased TREM2 signaling in CAM is desirable.

- *Regarding the relevance of TREM2 signaling in CAMs:* The present study shows that TREM2 signaling via DAP12 is increased in CAMs compared to microglia. From our snRNA seq analysis, CAMs have increased *Trem2* (1.20-fold) and *Tyrobp* (1.05-fold) as compared to homeostatic microglia, which is a slight albeit insignificant upregulation. However, we would argue that the importance of a gene's function for a certain cell type or state cannot solely be inferred from its expression level. Our prior work has shown that TREM2 function is not necessary for the expression of CAM markers such as CLEC7A (see Supplementary Figure 9G in Zarb et al., 2021 (PMID: 33637522)), but the loss of *Trem2* greatly increases the calcification load. Thus, while TREM2 function in microglia and CAMs is necessary, we still do not understand the exact role TREM2 signaling plays in CAMs or microglia in relation to calcification. Therefore, there is a need for further studies, such as the adult induced

microglia (ideally CAM) specific knock out of *Trem2/Tyrobp* in *Pdgfr^{ret/ret}* mice, to assess the exact role of TREM2 signaling in CAM function. We have now included a discussion into the role of TREM2 signaling in page 13, lines 348 – 356.

- Regarding other interacting partners to DAP12: We have now included a line in the discussion and the text reads: "While pDAP12 generally indicates active TREM2 signaling, it is important to note that DAP12 interacts with other receptors such as SIGLEC-H (PMID: 16293595), which is also expressed in microglia (PMID: 28836308)." See page 13 lines 348-51.

6. In regard to Figure 1D, the authors mention *Fmn1* explicitly in the text. However, *Fmn1* cannot be found in Figure 1D. Please clarify.

As a more general comment to Figure 1D: several of the annotations are unreadable. Therefore, the Figure should be improved such that (i) annotations are readable and (ii) it is clear which data point is annotated.

-Thank you for pointing this out. The Figure 1D has been modified and we hope that annotations are readable.

7. In Figure 1F, the authors annotate an immature microglia population („im_MG"). However, in the main text of the manuscript, this population is not discussed at all. Please clarify.

- We thank the reviewer for noticing this oversight. We have now commented on this population (page 7 lines 173-174).

8. In Figure 2A, the authors stain for CLEC7A as a marker for CAM. Surprisingly, the corresponding gene is not among the DEGs when comparing CAM and homeostatic microglia (see list of DEGs in supplementary Excel file). Nevertheless, *Clec7a* was among the DEGs in an RNA-seq analysis in the previous publication from the same group: Zarb et al, 2021. This should be explained.

- Indeed, *Clec7a* is not DEG, this is because the *Clec7a* transcript is not detected in the snRNA seq dataset. When comparing bulk vs sc/sn RNA seq DEG lists, often discrepancies are found in the DEG lists. This arises due to technical challenges/issues with sc/snRNA seq workflows. Sc/snRNA seq is inefficient in capturing mRNA within the cell/nucleus leading to "drop out" events where even moderately expressed genes are not detected. This results in discrepancies in DEG lists between bulk and sc/snRNA seq approaches and additionally between sc/snRNA seq datasets (PMID: 28821273). Recognizing the general awareness of this shortcoming, we have decided to forego a detailed discussion on missing transcripts.

However, while our current snRNAseq analysis of CAMs showed a high expression of *Spp1* (also upregulated gene in Zarb et al, 2021 in the calcified tissue), the protein product osteopontin could not be detected in Zarb et al., 2021 (PMID: 33637522). Therefore, we decided to perform the *in situ* hybridization analysis to clarify unambiguously whether CAMs express *Spp1* or not. We detect *Spp1* expression on CAMs, however, we do not detect the protein product inside CAMs. This is important information and clearly demonstrates that although osteopontin mRNA (*Spp1*) can be used to identify CAMs, osteopontin protein detection cannot be used. This finding is now included in the manuscript – Supplementary

Figure 1F, page 6, lines 142-143 and the results are discussed in Discussion (page 12-13, lines 327-332).

9. This reviewer agrees that the very high anti-TREM2 antibody amounts as reported in Figure 3A could be due to increased BBB permeability in the *Pdgfb*(ret/ret) mouse model. This would result in much higher antibody concentrations in the brain than would normally be possible if the BBB was intact. Is it therefore possible that CAM are exhausted after 8 weeks of treatment simply because they are being overactivated? This should be discussed. Performing snRNA-seq analysis upon chronic AL002a treatment could be quite informative.

- It is possible that the higher amounts of antibody in *Pdgfb*^{ret/ret} mice could potentially lead to TREM2 overstimulation. In addition, TREM2 activation could be beneficial in a scenario where *Trem2* is impaired (haploinsufficiency) or when microglia are less efficient (aging). This point is discussed – see page 15-16, line 409-414.

We agree that sn/scRNA seq analysis could potentially be informative to gain a deeper understanding of the effects of antibody treatment on microglia and CAMs. However, in terms of feasibility a suggested snRNA seq approach poses a considerable challenge. Owing to their smaller numbers compared to homeostatic microglia and issues with isolation, based on our experience, we need to analyse at least 6 (*Pdgfb*^{ret/ret} mice) per condition – AL002a treated, isotype treated and untreated mice. When including the *Pdgfb*^{ret/wt} controls, this number results in at least 32 mice which is not feasible for us to do because the extremely high cost of this experiment (>100 000 \$ - costs to cover mouse husbandry, library preparation, sequencing, labor costs etc.. Of note, in Switzerland this sum covers nearly 2 years salary of a PhD student.). In addition, as far as we are aware, no single cell/bulk-RNA sequencing data is available from chronically TREM2-activated microglia. Considering the interest in developing TREM2 activating antibodies that have better brain penetrance, we agree that these experiments would provide valuable information.

Nevertheless, to probe the effect of the antibody treatment on microglia, we performed a bulk RNA sequencing analysis of enriched microglia after 1 dose of treatment isolated from the entire cerebrum of the mouse brain (i.e., without olfactory bulbs and cerebellum). Briefly, Cd11b+ cells were isolated from three *Pdgfb*^{ret/ret} mice per condition – untreated, AL002a or isotype treated (60 mg/kg, 1 dose, 1 week post i.p.). RNA extracted from these cells were used for total RNA library preparation using the Illumina TruSeq Total RNA protocol by the Functional Genomics Centre Zurich. The libraries were sequenced using Illumina Novaseq X plus with a 10B flow cell. The fastq files were then processed for further downstream analysis (*Rebuttal Figure 1A*).

Rebuttal Figure 1. Bulk RNA sequencing of microglia enriched from *Pdgfr^{ret/ret}* mice. A) Workflow for bulk total RNA sequencing analysis from enriched microglia. B) Principal component analysis of microglia enriched from AL002a treated (in red), isotype treated (in green) and untreated (in blue) *Pdgfr^{ret/ret}* mice cerebrums.

Analysis of the results shows that treatment with either antibody induces changes in enriched microglia (samples 1-3: AL002a, sample 5-6: isotype) compared to untreated controls (sample 7-9) (*Rebuttal Figure 1B*). However, probing for differential gene expression analysis between these samples revealed few differentially expressed genes which did not associate with any annotated pathway (*Rebuttal Figure 2*).

Rebuttal Figure 2. Gene set enrichment analysis of genes upregulated by AL002a treatment alone. A) Comparison of genes differentially expressed in AL002a vs untreated (UT) microglia (in red), AL002a vs Isotype (in green) and Isotype vs untreated (in yellow). The 34 genes unique to AL002a treatment were used for GSEA. B) The 34 genes unique to AL002a treatment do not point to any specific altered pathways in microglia.

10. On top of page 9, the authors mention that isotype-A647 colocalized with IBA1 positive microglia. Why does the isotype antibody colocalize with microglia? This is difficult to understand and should be explained.

- It is possible that the colocalization of the isotype antibody with microglia is due to the engagement of the Fc domain with Fcγ receptors expressed by microglia (see *Rebuttal Figure 3*). This is clarified in text page 9, lines 228-229.

Rebuttal Figure 3. Expression of Fc gamma receptors in microglia from our snRNA seq dataset.

11. In Figure 5D, this reviewer finds APOE stainings not convincing enough to conclude that APOE level is not altered upon AL002a treatment. Do the authors have quantifications to support their conclusion?

- We agree with the reviewer that for solid conclusion the APOE expression should have been quantified. In the revised manuscript we present this data. The quantification revealed that TREM2 activation with AL002a increased resulted higher APOE expression in CAMs compared to IgG treated although the difference was not statistically significant. This data has been added (Fig 5E) and the text modified accordingly (page 11, lines 298-299). In addition, we also performed quantification of the CLEC7A expression in CAMs after control IgG and AL002a treatment, which confirmed that the expression was unaltered by AL002a treatment compared to isotype IgG. This quantification is now included in Figure 5F and the text modified accordingly (result: page 11, lines 295-298, discussion: page 15, line 404-408).

12. Which brain region was analyzed in Figure 3A? Is this whole brain? This needs to be mentioned.

- The data presented in Figure 3A was obtained from cerebrum. This is now stated in the Figure 3 legend.

13. Are the images shown in Figure 3C taken from the deep brain region? Please clarify.

- Yes, the images were acquired from microglia in the deep brain region and is now added to the figure legend for Figure 3C.

Minor points

1. In the beginning of the Introduction, the authors talk about „specific conditions" and „pediatric disorders" (toward end of page 2). This needs to be more concrete.

- Thank you for pointing this awkward sentence out. We have modified the text and it now reads as follows: “*These radiological features are also characteristic of certain pediatric conditions, and the pattern of calcification can aid in their diagnosis.*” (page 2 lines: 43-45) We would like to avoid specifying any particular condition because the approaches that aid diagnoses are rather complex and the list of conditions is rather long. The references contain all necessary information for those seeking additional information.

2. On page 6, it should be „Selplg" instead of „Selpg".

- This is corrected.

3. On page 6, what do the authors mean by „genome transcriptomes"?

- Thank you for pointing this out. The sentence reads now *“To address this, we integrated deep brain microglial nuclei from our dataset with cortical microglia nuclei from the Zhou et al. snRNA-seq dataset (Zhou Y et al, 2020) and compared the transcriptomes of CAMs and DAMs.”*(page 7, lines 163-165).

4. On page 7, what is the difference between „microglia" and „non-CAMs"? The authors should be consistent in regard to nomenclature.

- We have deleted “non-CAMs”.

5. On page 8, the authors mention two times that the isotype antibody was not detected in the immunoassay which detects the anti-TREM2 antibody (i.e. AL002a). Mentioning it once is enough.

- We have modified the text and removed the repetition.

6. In Figures 2 and 5, it would be desirable to show separate IBA1/OPN stainings.

- We are unsure if we have correctly interpreted your comment regarding IBA1/OPN staining. The images in Figure 2 do not contain OPN staining. Figure 5C, is the only figure where this combination is shown. We have now included a panel showing IBA1+OPN in Figure 5C and IBA1+DAP1 in Figure 5D.

7. Please add mouse age and mouse N to Figure 1 legend.

- This information has now been added

8. Please add mouse age to Figure 2 legend.

- This information has now been added

9. Figure 3 would benefit from a panel showing a scheme of the chronic treatment paradigm.

- The chronic treatment paradigm scheme has been added as a Supplementary Figure 2A.

10. Please add mouse age to legend of Figures 3A-C.

- This information has now been added

11. In Figure 3G, the error bars are most likely SEM and not SD (in the Materials and Methods the authors state that all Figures show SD).

- We are grateful to the reviewer for catching this error. We have replaced Figure 3G with a new graph indicating SD.

12. In Figure 3, it seems that data as shown in panels A-C are derived from a single antibody dose study while the data in panels D-H are derived from the multiple dose study. This should be clarified in the figure legend.

- This has now been clarified.

13. In Figure 4B, it seems that oxygen (O) content in calcifications in AL002a treated mice is lower compared to isotype treated mice. Please comment.

- While EDX analysis can indicate the presence of elements, it is less reliable for quantitative comparisons, especially in SEM. The variation in oxygen signal between AL002a and isotype groups could be due to factors like the composition of underlying material or differences in signal filtering. Therefore, a definitive conclusion about oxygen content differences cannot be drawn from this data alone.

14. In the legend of Figure 5B one can find only one t-test-associated p value. Is it the same for both comparisons („Ki67+ microglia" and „all microglia") or is one p value missing? Please clarify.

- We thank the reviewer for bringing our attention to this. One p value was indeed missing and is now added to the Figure 5 legend, page 25.

October 10, 2024

RE: Life Science Alliance Manuscript #LSA-2024-02796R

Dr. Annika Keller
University Hospital of Zurich
Division of Neurosurgery, Zürich University
Wagistrasse 12
Zürich CH-8952
Switzerland

Dear Dr. Keller,

Thank you for submitting your revised manuscript entitled "Targeting TREM2 signaling shows limited impact on cerebrovascular calcification". We would be happy to publish your paper in Life Science Alliance pending final revisions necessary to meet our formatting guidelines.

- please address Reviewer 1's remaining comment
- please be sure that the authorship listing and order is correct
- please add an Abstract to our system
- please add the Twitter handle of your host institute/organization as well as your own or/and one of the authors in our system
- please add your supplementary figure legends to the main figure legend section
- please upload your tables as separate editable doc or excel files
- the snRNA-seq data should be made publicly accessible at this point, removing the need for the Reviewer access token in the Data Availability statement

Figure Check:

- please add scale bars to Figure S1A

LSA now encourages authors to provide a 30-60 second video where the study is briefly explained. We will use these videos on social media to promote the published paper and the presenting author (for examples, see <https://docs.google.com/document/d/1-UWCfbE4pGcDdcgzcmiuJl2XMBJnxKYeqRvLLrLS08s/edit?usp=sharing>). Corresponding or first-authors are welcome to submit the video. Please submit only one video per manuscript. The video can be emailed to contact@life-science-alliance.org

A. FINAL FILES:

B. MANUSCRIPT ORGANIZATION AND FORMATTING:

Sincerely,

Reviewer #1 (Comments to the Authors (Required)):

All of the points which I raised during the initial review of the manuscript from Sridhar et al have been carefully addressed. I congratulate the authors to a nice manuscript and thus recommend its publication in LSA. The only thing which should still be addressed is to further improve Fig. 1D. It has been substantially improved since its original submission but for several annotations it is still unclear to which of the dots they belong. The authors may want to use arrowheads to make clear which of the dots they are annotating with the respective gene. Once that has been done this article should be published.

Best regards

We thank you for accepting our revised manuscript Sridhar et al., "*Targeting TREM2 signaling shows limited impact on cerebrovascular calcification*" for publication.

We have made the following changes required to meet the formatting guidelines of *Life Science Alliance*.

-please address Reviewer 1's remaining comment:

Reviewer #1 (Comments to the Authors (Required)):

All of the points which I raised during the initial review of the manuscript from Sridhar et al have been carefully addressed. I congratulate the authors to a nice manuscript and thus recommend its publication in LSA. The only thing which should still be addressed is to further improve Fig. 1D. It has been substantially improved since its original submission but for several annotations it is still unclear to which of the dots they belong. The authors may want to use arrowheads to make clear which of the dots they are annotating with the respective gene.

Once that has been done this article should be published.

Answer: We have now modified Fig. 1D to increase the point size of the annotated genes and use arrows to indicate the exact dots.

-please be sure that the authorship listing and order is correct: *The authorship listing and order is correct.*

-please add an Abstract to our system: *The abstract has now been added to the system*

-please add the Twitter handle of your host institute/organization as well as your own or/and one of the authors in our system: *a short text highlighting our work and the twitter handle for our lab are now included.*

-please add your supplementary figure legends to the main figure legend section: *The supplementary figure legends have now been moved to the main figure legend section.*

-please upload your tables as separate editable doc or excel files: *Source data files and supplementary tables 1 and 2 are now uploaded as separate files.*

-the snRNA-seq data should be made publicly accessible at this point, removing the need for the Reviewer access token in the Data Availability statement: *The snRNA-seq data is now public and the reviewer access token has been removed from the Data Availability statement.*

Figure Check:

-please add scale bars to Figure S1A:

Answer: The two images in Figure S1A were generated from the blue brain cell atlas portal <https://bbp.epfl.ch/nexus/cell-atlas/> which does not provide scale bars. We added the following sentence to the figure legend "Images depicting brain regions were generated from <https://bbp.epfl.ch/nexus/cell-atlas/>."

A. FINAL FILES:

-- An editable version of the final text (.DOC or .DOCX) is needed for copyediting (no

Manuscript # LSA-2024-02796R

PDFs). The final manuscript has been uploaded as an editable .docx file.

-- High-resolution figure, supplementary figure and video files uploaded as individual files. See our detailed guidelines for preparing your production-ready images, <https://www.life-science-alliance.org/authors>. : High resolution figures have been uploaded as pdf files.

-- Summary blurb (enter in submission system): A short text summarizing in a single sentence the study (max. 200 characters including spaces). This text is used in conjunction with the titles of papers, hence should be informative and complementary to the title. It should describe the context and significance of the findings for a general readership; it should be written in the present tense and refer to the work in the third person. Author names should not be mentioned.: Summary blurb has been uploaded to the system.

B. MANUSCRIPT ORGANIZATION AND FORMATTING:

The manuscript has been organized and formatted according to LSA requirements.

October 16, 2024

RE: Life Science Alliance Manuscript #LSA-2024-02796RR

Dr. Annika Keller
University of Zurich
Dept. of Neurosurgery
Wagistrasse 12
Zürich CH-8952
Switzerland

Dear Dr. Keller,

Thank you for submitting your Research Article entitled "Targeting TREM2 signaling shows limited impact on cerebrovascular calcification". It is a pleasure to let you know that your manuscript is now accepted for publication in Life Science Alliance. Congratulations on this interesting work.

DISTRIBUTION OF MATERIALS:

Again, congratulations on a very nice paper. I hope you found the review process to be constructive and are pleased with how the manuscript was handled editorially. We look forward to future exciting submissions from your lab.

Sincerely,
